# HIV-1 uncoating by release of viral cDNA from capsid-like structures in the nucleus of infected cells

Thorsten G Müller[1], Vojtech Zila[1], Kyra Peters[1], Sandra Schifferdecker[1], Mia Stanic[2], Bojana Lucic[2], Vibor Laketa[1,3], Marina Lusic[2,3], Barbara Müller[1], Hans-Georg Kräusslich[1,3]*

[1]Department of Infectious Diseases Virology, University Hospital Heidelberg, Heidelberg, Germany; [2]Department of Infectious Diseases Integrative Virology, University Hospital Heidelberg, Heidelberg, Germany; [3]German Center for Infection Research, partner site Heidelberg, Heidelberg, Germany

**Abstract** HIV-1 replication commences inside the cone-shaped viral capsid, but timing, localization, and mechanism of uncoating are under debate. We adapted a strategy to visualize individual reverse-transcribed HIV-1 cDNA molecules and their association with viral and cellular proteins using fluorescence and correlative-light-and-electron-microscopy (CLEM). We specifically detected HIV-1 cDNA inside nuclei, but not in the cytoplasm. Nuclear cDNA initially co-localized with a fluorescent integrase fusion (IN-FP) and the viral CA (capsid) protein, but cDNA-punctae separated from IN-FP/CA over time. This phenotype was conserved in primary HIV-1 target cells, with nuclear HIV-1 complexes exhibiting strong CA-signals in all cell types. CLEM revealed cone-shaped HIV-1 capsid-like structures and apparently broken capsid-remnants at the position of IN-FP signals and elongated chromatin-like structures in the position of viral cDNA punctae lacking IN-FP. Our data argue for nuclear uncoating by physical disruption rather than cooperative disassembly of the CA-lattice, followed by physical separation from the pre-integration complex.

*For correspondence:
hans-georg.kraeusslich@med.uni-heidelberg.de

Competing interests: The authors declare that no competing interests exist.

## Introduction

Retroviral replication involves reverse transcription of the viral RNA genome and requires nuclear entry of the subviral complex to allow for chromosomal integration of the viral cDNA mediated by the viral integrase (IN) (*Lusic and Siliciano, 2017*). Although the gammaretrovirus murine leukemia virus requires nuclear envelope breakdown during mitosis for productive replication, HIV-1 and other lentiviruses infect non-dividing cells, implying that the subviral complex can pass through the intact nuclear envelope (*Suzuki and Craigie, 2007*). Reverse transcription mediated by the viral reverse transcriptase (RT) is initiated in the cytoplasm, but recent evidence indicates that cDNA synthesis is completed inside the nucleus (*Burdick et al., 2020*; *Dharan et al., 2020*; *Selyutina et al., 2020*), at least in the case of HIV-1. The cytoplasm is a hostile environment for retroviral genome replication: exposure of cytoplasmic DNA to cellular nucleic acid sensors would lead to induction of innate immunity (*Doitsh et al., 2014*; *Monroe et al., 2014*), thereby aborting viral infection. The viral cone-shaped capsid apparently plays a central role in guiding and shielding (*Rasaiyaah et al., 2013*; *Sumner et al., 2020*) the genome through the cytosolic environment (*Campbell and Hope, 2015*; *Novikova et al., 2019*). It consists of ~1200–1500 CA molecules assembled into hexamers and pentamers (*Briggs et al., 2003*), which have been shown to interact with components of the nuclear pore complex (NPC) (*Bhattacharya et al., 2014*; *Matreyek et al., 2013*; *Price et al., 2014*), implying a role for the CA-lattice in nuclear entry (*Di Nunzio et al., 2013*; *Lelek et al., 2015*; *Matreyek et al., 2013*). However, the HIV-1 capsid, with a length of ~120 nm and a width of ~60 nm

**eLife digest** When viruses infect human cells, they hijack the cell's machinery to produce the proteins they need to replicate. Retroviruses like HIV-1 do this by entering the nucleus and inserting their genetic information into the genome of the infected cell. This requires HIV-1 to convert its genetic material into DNA, which is then released from the protective shell surrounding it (known as the capsid) via a process called uncoating.

The nucleus is enclosed within an envelope containing pores that molecules up to a certain size can pass through. Until recently these pores were thought to be smaller than the viral capsid, which led scientists to believe that the HIV-1 genome must shed this coat before penetrating the nucleus. However, recent studies have found evidence for HIV-1 capsid proteins and capsid structures inside the nucleus of some infected cells. This suggests that the capsid may not be removed before nuclear entry or that it may even play a role in helping the virus get inside the nucleus.

To investigate this further, Müller et al. attached fluorescent labels to the newly made DNA of HIV-1 and some viral and cellular proteins. Powerful microscopy tools were then used to monitor the uncoating process in various cells that had been infected with the virus. Müller et al. found large amounts of capsid protein inside the nuclei of all the infected cells studied. During the earlier stages of infection, the capsid proteins were mostly associated with viral DNA and the capsid structure appeared largely intact. At later time points, the capsid structure had been broken down and the viral DNA molecules were gradually separating themselves from these remnants.

These findings suggest that the HIV-1 capsid helps the virus get inside the nucleus and may protect its genetic material during conversion into DNA until right before integration into the cell's genome. Further experiments studying this process could lead to new therapeutic approaches that target the capsid as a way to prevent or treat HIV-1.

at its wide end (*Mattei et al., 2016*), is presumed to exceed the dimensions of the NPC channel with a reported maximal diameter of ~40 nm (*von Appen et al., 2015*). This implies that capsid uncoating should occur – at least partially – prior to nuclear entry, and various publications reported uncoating either in the cytoplasm (*Cosnefroy et al., 2016*; *Hulme et al., 2011*; *Mamede et al., 2017*; *Xu et al., 2013*) or at the nuclear pore (*Arhel et al., 2007*; *Burdick et al., 2017*; *Francis and Melikyan, 2018*), with some evidence for cell-type-dependent differences. On the other hand, nuclear HIV-1 pre-integration complexes (PIC) were found to retain varying amounts of CA molecules (*Bejarano et al., 2019*; *Burdick et al., 2020*; *Chin et al., 2015*; *Hulme et al., 2015*; *Stultz et al., 2017*; *Zila et al., 2019*), at least in certain cell types (*Zila et al., 2019*), and recent reports indicating the presence of intact capsid lattice (*Dharan et al., 2020*; *Selyutina et al., 2020*) and capsid-like structures (*Zila et al., 2021*) inside the nucleus challenged the current models of early HIV-1 replication. Accordingly, the timing, subcellular localization, trigger and mechanism of HIV-1 capsid uncoating are still under debate.

Studying early HIV-1 replication is hampered by the fact that most cytoplasmic entry events appear to be non-productive in tissue culture (*Klasse, 2015*; *Sanjuán, 2018*). Therefore, characterization of individual subviral complexes containing viral cDNA with respect to their content, subcellular distribution and trafficking is required to shed light on the pathway of productive replication. Viral cDNA can be visualized in fixed cells using fluorescence in situ hybridization (FISH) (*Marini et al., 2015*) or its derivatives using branched probes (*Chin et al., 2015*; *Puray-Chavez et al., 2017*), but the harsh assay conditions destroy the native cellular environment and impair immunofluorescence analysis. Incorporation of the modified nucleoside 5-ethynyl-2′-deoxyuridine (EdU) allowed the detection of actively reverse transcribing HIV-1 complexes by visualizing de novo synthesized viral DNA via click chemistry (*Peng et al., 2015*; *Stultz et al., 2017*), but this approach is also limited to fixed cells and cellular extraction precludes high-resolution analysis (*Müller et al., 2019*). To overcome these limitations, we adapted a live cell compatible genetically encoded system (ANCHOR) that allows single-molecule gene labeling (*Germier et al., 2017*; *Saad et al., 2014*), and has been applied for visualization of DNA from Adenovirus (*Komatsu et al., 2018*), Cytomegalovirus (*Mariamé et al., 2018*), and HIV-1 (*Blanco-Rodriguez et al., 2020*). This system is based on the prokaryotic chromosomal partitioning system ParB-parS, where ParB

(designated OR) specifically binds the parS seed sequence (designated ANCH). Multiple copies of parS introduced into the HIV-1 genome act as nucleation sites to oligomerize the fluorescently labeled OR protein when the reverse transcribed ANCH cDNA sequence becomes accessible to the fusion protein.

Here, we show that HIV-1 cDNA containing subviral complexes associated with CA are detected in the nucleus of infected cells, including primary CD4$^+$ T cells. Over time, cDNA containing complexes segregate from CA and the bulk of viral replication proteins, confirming nuclear uncoating. Using 3D correlative light and electron microscopy (CLEM), we detected capsid-like structures at the position of nuclear IN-FP-containing complexes, whereas elongated, chromatin-like densities were observed at the position of viral cDNA punctae. Importantly, strong CA signals were observed on nuclear HIV-1 complexes in all cell types analyzed, indicating that prior failure to detect nuclear CA by immunofluorescence was largely due to masked epitopes.

## Results

### The ANCHOR system enables visualization of integrated and unintegrated HIV cDNA in the nucleus of infected cells

To test for retention of the ANCH sequence and efficiency of visualizing HIV-1 cDNA following reverse transcription, we stably transduced HeLa-based TZM-bl cells with different amounts of an ANCH3 containing HIV-1-based vector. These cell populations were subsequently transduced or transfected with an expression vector for eGFP.OR3 (*Figure 1—figure supplement 1a*). Confocal microscopy revealed distinct eGFP.OR3 punctae in the nuclei of >90% of transfected cells (*Figure 1—figure supplement 1b–d*). At low multiplicities of transduction with the ANCH-vector, where the majority of cells is expected to originate from a single integration event, we observed an average of $1.6 \pm 0.29$ and $1.4 \pm 0.34$ eGFP.OR3 punctae per nucleus (*Figure 1—figure supplement 1d*). The number of punctae correlated with the multiplicity of transduction over a wide range (*Figure 1—figure supplement 1d*). Of note, the eGFP.OR3 signal was stable for more than 4 weeks (when unintegrated viral cDNA species are expected to be degraded) suggesting that integrated viral DNA can be detected. Thus, the ANCH sequence is retained during reverse transcription, and this approach can be used to detect HIV-1 cDNA during the early replication phase.

Next, we introduced the ~1,000 bp ANCH3 sequence into the HIV-1 proviral plasmid pNLC4-3 (*Bohne and Kräusslich, 2004*) (HIV ANCH) replacing part of the *env* gene (*Figure 1a*). Virus-like particles were pseudotyped with the vesicular stomatitis virus G protein (VSV-G) or HIV-1 Env as indicated. They also contained exogenously expressed IN tagged with a fluorescent marker (IN-FP) (*Albanese et al., 2008*), in addition to wild-type IN encoded by the virus, for visualization of subviral replication complexes. These particles were used to infect polyclonal TZM-bl cell populations stably transduced to express OR3 fused with either eGFP, mScarlet (*Bindels et al., 2017*) or the stainable SNAP-tag (*Keppler et al., 2003*); these cells also stably expressed fluorescently tagged Lamin B1 (LMNB1) to clearly distinguish nuclear and cytoplasmic events. *Figure 1b* shows TZM-bl eBFP2. LMNB1 and eGFP.OR3 expressing cells infected with HIV ANCH. Distinct infection-induced eGFP punctae were clearly detected in the nuclei of these cells, but were not observed in the cytoplasm where eGFP.OR3 was diffusely distributed (*Figure 1b*). Distinct nuclear eGFP punctae were not detected in uninfected cells (*Figure 1—figure supplement 2a*).

To determine whether HIV ANCH cDNA became chromosomally integrated, we infected the human SupT1 T-cell line with HIV ANCH and analyzed the copy number of integrated proviral genomes by semi-quantitative Alu-PCR; this experiment could not be performed in TZM-bl cells, since these cells carry multiple HIV-1 LTR copies from prior lentiviral vector transductions. Integrated proviral DNA was readily detected in HIV ANCH infected SupT1 cells, but was not observed when infection was performed in the presence of an RT- or IN-inhibitor (*Figure 1c*). Similar to TZM-bl cells, SupT1 cells also showed nuclear eGFP.OR3 punctae following infection with HIV ANCH (see below, *Figure 7—figure supplement 1a*). We then determined the number of eGFP.OR3 punctae in TZM-bl cells using confocal microscopy of cells fixed at different time points after infection with HIV ANCH (*Figure 1d*) and performed parallel quantitation of total HIV-1 cDNA and 2-LTR circles (representing unintegrated nuclear HIV-1 cDNA) using digital droplet PCR (ddPCR) (*Figure 1e*). Total cDNA levels became saturated at 10 hr post infection (h p.i.) and 2-LTR circles peaked at 24 h p.i.;

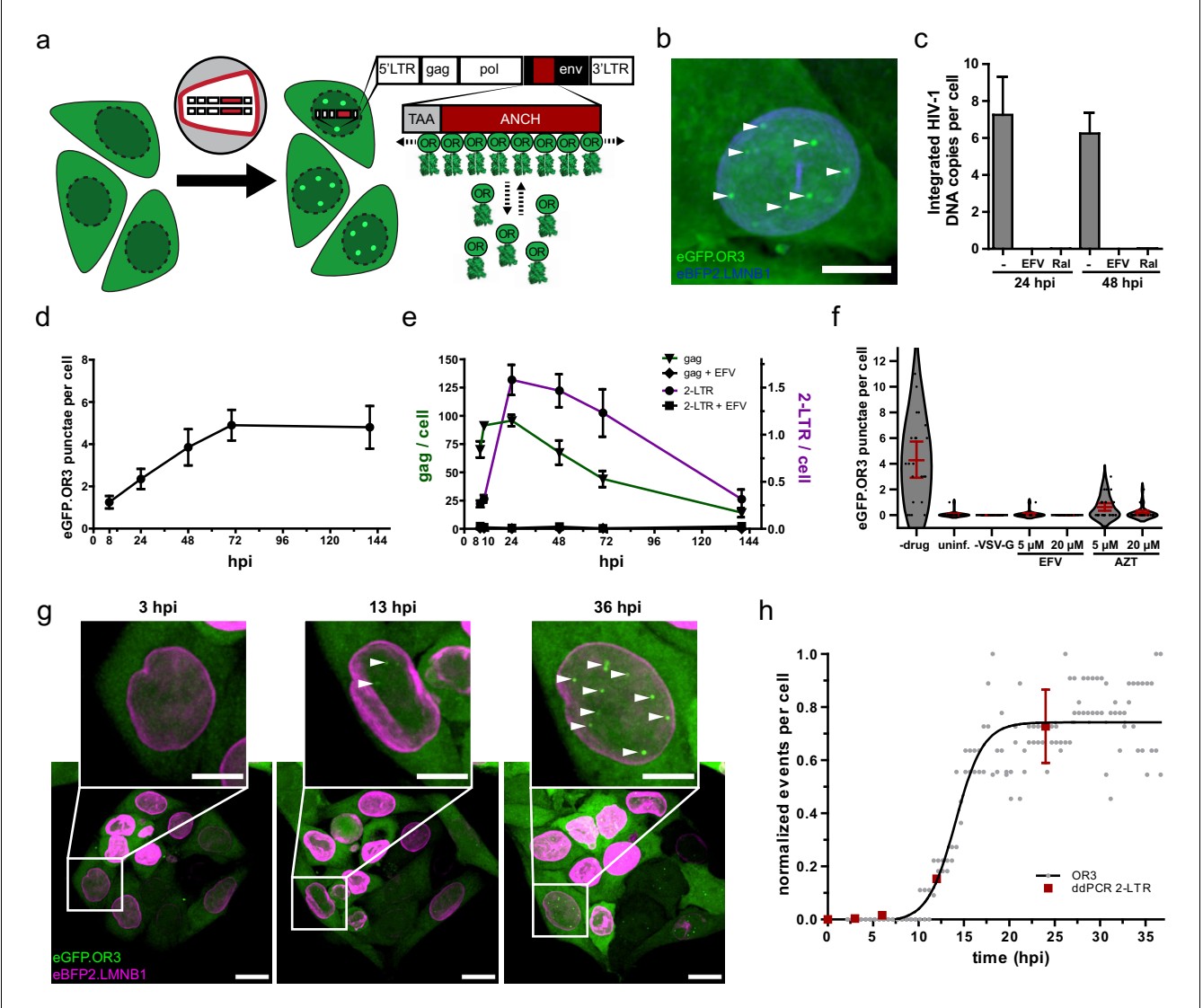

**Figure 1.** Visualization of HIV-1 dsDNA within the nucleus of infected cells. (a) Scheme of the ANCHOR dsDNA visualization system. Fluorescently tagged OR3 binds to the ANCH sequence on the viral dsDNA (see *Figure 1—figure supplement 1*). (b) eGFP.OR3 punctae detected in the nuclei of infected cells. TZM-bl eBFP2.LMNB1 eGFP.OR3 cells were infected with VSV-G pseudotyped HIV-1$_{NL4-3}$ ANCH (30 μUnits RT/cell, MOI 6), fixed at 55 h p.i. and imaged using SDCM. Six independent experiments with two independent virus preparations were performed. Maximum intensity projection (MIP) of a representative cell is shown. Note that some cells show one or two cytoplasmic, mostly perinuclear, infection-independent accumulations of the OR3 fusion protein, which we identified as multivesicular bodies (MVB) by CLEM-ET (*Figure 1—figure supplement 2*). Scale bar: 5 μm. (c) Quantification of integrated HIV-1$_{NL4-3}$ ANCH provirus using nested Alu-LTR PCR. SupT1 cells were infected using VSV-G pseudotyped HIV ANCH (10 μU RT/cell, corresponding to an MOI ~ 8 under the conditions used). Raltegravir (RAL) or Efavirenz (EFV) were added at the time of infection as indicated. Data are shown for one experiment performed in biological triplicates; error bars represent SD. (d,e) eGFP.OR3 punctae are stable for more than 6 days while total and unintegrated DNA declined (see *Figure 1—figure supplement 3*). Quantification of eGFP.OR3 punctae by SDCM (d), and gag and 2-LTR cDNA by ddPCR (e) over time within the same experiment. Infection was performed as in (b). (d) Data from one of two independent experiments (n = 20 cells per time point, error bars represent SEM) (e) Data from one experiment performed in biological triplicates (error bars represent SEM). (f) Detection of eGFP.OR3 punctae is HIV-1 cell fusion and reverse transcription dependent. TZM-bl eGFP.OR3 cells were infected with VSV-G pseudotyped NNHIV ANCH (10 μU RT/cell) and imaged under live conditions at 27 h p.i. Control experiments were performed adding HIV-1 RT inhibitors EFV or azidothymidine (AZT) at time of infection or using particles lacking a fusion protein ('-VSV-G'). n = 20–29 cells were analyzed per sample and error bars represent 95 % CI; The graph shows data from one of three independent experiments (see *Figure 1—figure supplement 4*). (g, h) Live cell imaging of vDNA punctae formation. TZM-bl eBFP2.LMNB1 and eGFP.OR3 cells were infected with NNHIV ANCH (30 μU RT/cell) and image acquisition by SDCM was initiated at 2 h p.i. 3D stacks were acquired every 30 min for 36 hr. Representative data from one of four independent experiments are shown. MIP of a representative cell is shown. Scale bars: 10 μm (overview), 5 μm (enlargement). See *Figure 1—video 1*. (h) Quantification of eGFP.OR3 punctae formation in cells from video shown in (g). Analyzed was the eGFP.OR3 punctae formation from two cells within

*Figure 1 continued on next page*

*Figure 1 continued*

the field of view (FOV) with each point representing the normalized amount of OR3 punctae per nucleus at the respective timepoint. Data were fit to a logistic growth model giving $t_{1/2}$ = 14.1 ± 0.5 h p.i. Detection of 2-LTR circles (mean and SEM) by ddPCR in biological triplicates is shown for TZM-bl cells infected with NNHIV WT.

The online version of this article includes the following video, source data, and figure supplement(s) for figure 1:

**Source data 1.** Data corresponds to the number of eGFP.OR3 punctae per cell nucleus (*Figure 1f*), the normalized number of eGFP.OR3 punctae per cell nucleus (*Figure 1h*) and normalized number of 2-LTR circles quantified by ddPCR in biological triplicates (*Figure 1h*).

**Figure supplement 1.** The ANCHOR system enables sensitive visualization of integrated lentiviral DNA without loss through recombination during reverse transcription.

**Figure supplement 1—source data 1.** Data corresponds to the number of cells showing between 0 and 4 eGFP.OR3 punctae per cell nucleus (*Figure 1—figure supplement 1c*) and the number of eGFP.OR3 punctae per cell nucleus for different MOIs (*Figure 1—figure supplement 1d*).

**Figure supplement 2.** CLEM-ET analysis of infection-independent cytoplasmic OR3 fusion protein accumulations.

**Figure supplement 3.** eGFP.OR3 punctae in infected cell nuclei are detected independent of HIV-1 transcription.

**Figure supplement 3—source data 1.** Data corresponds to the number of IN.SNAP.SiR or eGFP.OR3 punctae per cell nucleus with and without flavopiridol treatment (*Figure 1—figure supplement 2a*).

**Figure supplement 4.** Representative images from control experiments.

**Figure 1—video 1.** Live cell imaging of viral DNA punctae formation in TZM-bl cells.

https://elifesciences.org/articles/64776#fig1video1

---

both species strongly declined over the following 5 days (*Figure 1e*). eGFP punctae, on the other hand, increased over the first 72 hr, but then remained stable over the following 3 days despite the observed loss of HIV-1 cDNA species (*Figure 1d*). Taken together, these results clearly indicated that integrated HIV-1 DNA can be detected using the ANCHOR system. To determine whether detection of integrated proviral copies may be influenced by RNA transcription at the respective site, we compared the number of eGFP.OR3 punctae in HIV ANCH infected TZM-bl cells treated with the CDK9/p-TEFb inhibitor Flavopiridol or solvent control. No difference was observed (*Figure 1—figure supplement 3a–c*), suggesting that the dynamic nature of OR3 recruitment does not interfere with transcription.

Next, we generated a non-infectious derivative of HIV ANCH termed NNHIV ANCH to allow for live cell imaging outside the BSL3 facility. NNHIV ANCH is based on the previously reported plasmid NNHIV that carries point mutations in the active site of IN and a deletion in the *tat* gene ($IN_{D64N/D116N}$ $tat_{\Delta33-64bp}$) (*Zila et al., 2021*). This derivative retains reverse transcription and nuclear import ability, whereas integration and transcription are blocked. TZM-bl eGFP.OR3 cells infected with VSV-G pseudotyped NNHIV ANCH also showed nuclear eGFP.OR3 punctae (*Figure 1f–h*) indicating that unintegrated HIV-1 cDNA is detected by the ANCHOR system as well. eGFP.OR3 punctae were not detected when cells were treated with NNHIV ANCH lacking VSV-G and were absent or strongly reduced in the presence of the RT inhibitors efavirenz (EFV) or azidothymidine (AZT) (*Figure 1f* and *Figure 1—figure supplement 4*).

To investigate the dynamics of appearance of eGFP.OR3 punctae in NNHIV ANCH infected TZM-bl cells, we performed live cell imaging experiments using spinning disk confocal microscopy (SDCM). The onset of marker recruitment to viral cDNA in the nucleus was observed at 7–8 h p.i., while the half-maximal signal was reached between 13 and 15 h p.i. (*Figure 1g,h*; *Figure 1—video 1*). Again, no infection-induced eGFP.OR3 punctae were detected in the cytosol of infected cells. The onset of nuclear HIV-1 2-LTR detection using ddPCR coincided with the appearance of eGFP.OR3 punctae (*Figure 1h*). Formation of both 2-LTR circles (dependent on nuclear NHEJ components and ligase IV [*Li et al., 2001*]) and eGFP.OR3 punctae requires viral cDNA to be synthesized and accessible to proteins not present in the subviral replication complex. Accordingly, the lack of cytoplasmic eGFP.OR3 punctae may be due to incomplete cDNA synthesis in the cytoplasm and/or to shielding of the cDNA from the fusion protein until full capsid uncoating in the nucleus. To address this question, we focused on the timing and quantification of reverse transcription in the described system.

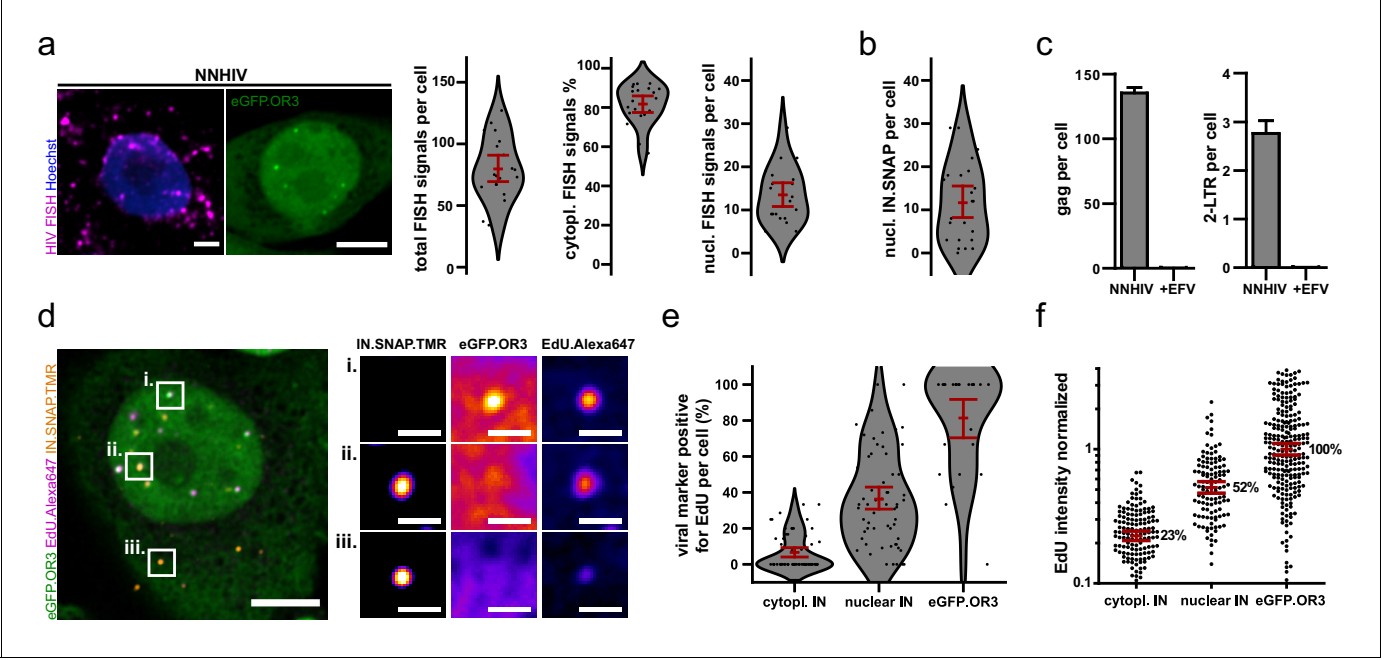

**Figure 2.** Viral cDNA in cytosolic complexes does not recruit OR3 proteins and contains less DNA compared to nuclear complexes. (**a**) HIV FISH staining of TZM-bl cells infected with VSV-G pseudotyped NNHIV ANCH (10 μU/RT cell; 24 h p.i.). A cell showing nuclear eGFP.OR3 signals from a different experiment is shown for comparison (two left panels). Scale bars: 5 μm. Quantification of the total number of FISH signals per cell, the percentage of cytoplasmic FISH signals and the total number of nuclear FISH signals per cell (right panels). The number of background nuclear FISH signals in uninfected cells (from prior HIV LTR integrations in the TZM-bl cell line) was subtracted from infected cells. Data points represent individual cells and error bars 95 % CI. (**b**) Quantification of nuclear IN.SNAP signals per cell. TZM-bl eBFP2.LMNB1 eGFP.OR3 cells were infected with VSV-G pseudotyped NNHIV ANCH (10 μU/RT cell; 24 h p.i.). Data points represent individual cells and error bars 95 % CI. (**c**) ddPCR quantification of late RT products (gag region; left panel) and 2-LTR circles (right panel). TZM-bl cells were infected using the same amount of VSV-G pseudotyped NNHIV ANCH (10 μU RT/cell, 24 h p.i.) as in *Figure 1f*, (**a** and **b**). Mean and SEM of one experiment performed in biological triplicates are shown. (**d**) TZM-bl eBFP2.LMNB1 and eGFP.OR3 expressing cells were infected with VSV-G pseudotyped and IN.SNAP.TMR labeled NNHIV ANCH (30 μU RT/cell) in the presence of EdU. Cellular DNA synthesis was blocked by aphidicolin (APC). At 24 h p.i. cells were fixed and click labeled. Two independent experiments were performed in biological triplicates. A 2 μm MIP of a representative cell and three enlarged single z slices are shown. See *Figure 2— figure supplement 1* for more examples. Scale bars: 5 μm (MIP) and 1 μm (enlargements). (**e,f**) Quantification of data from the experiment described in (**d**). Pooled data from one experiment performed in biological triplicates are shown, with data points representing individual cells (**e**) or subviral complexes (**f**); error bars represent 95 % CI. (**e**) Percentage of EdU-positive viral marker spots (IN or OR3) per cell. (**f**) Intensity of EdU signals associated with the respective viral marker. Data were normalized to the mean signal of eGFP.OR3 punctae. Differences are statistically significant (p<0.0001; two-tailed Student's t-test).

The online version of this article includes the following source data and figure supplement(s) for figure 2:

**Source data 1.** Data corresponds to the total FISH signals per cell, percent of cytoplasmic FISH signal, number of nuclear FISH signals per cell (*Figure 2a*) and number of nuclear IN.SNAP signals per cell (*Figure 2b*), the percentages of EdU-positive subviral structures per cell (*Figure 2e*) and the normalized (to the geometrical mean EdU intensity of eGFP.OR3 objects) intensities of EdU per cell localized at different subviral structures (*Figure 2f*).

**Figure supplement 1.** Exemplary images of newly synthesized DNA content in subviral complexes.

### Nuclear eGFP.OR3 punctae contain higher amounts of de novo synthesized HIV-1 cDNA than nuclear IN-positive complexes not recruiting eGFP.OR3

Quantitation of reverse transcription products was performed using FISH analysis. A large proportion (~80%) of RT products detected by FISH was localized in the cytoplasm of infected cells, whereas no eGFP.OR3 punctae were observed in this compartment (*Figure 2a*; note that the harsh sample treatment required for FISH detection did not preserve eGFP.OR3 punctae, precluding analysis of both markers in the same specimen). The number of nuclear FISH signals per cell (~14) (*Figure 2a*, right) was similar to the number of nuclear IN.SNAP signals (~12) (*Figure 2b*). A similar distribution of late RT products was also observed by ddPCR analysis under the same infection conditions: ~130 *gag* reverse transcripts and 2–3 2-LTR circles were detected per cell (*Figure 2c*); the slightly lower

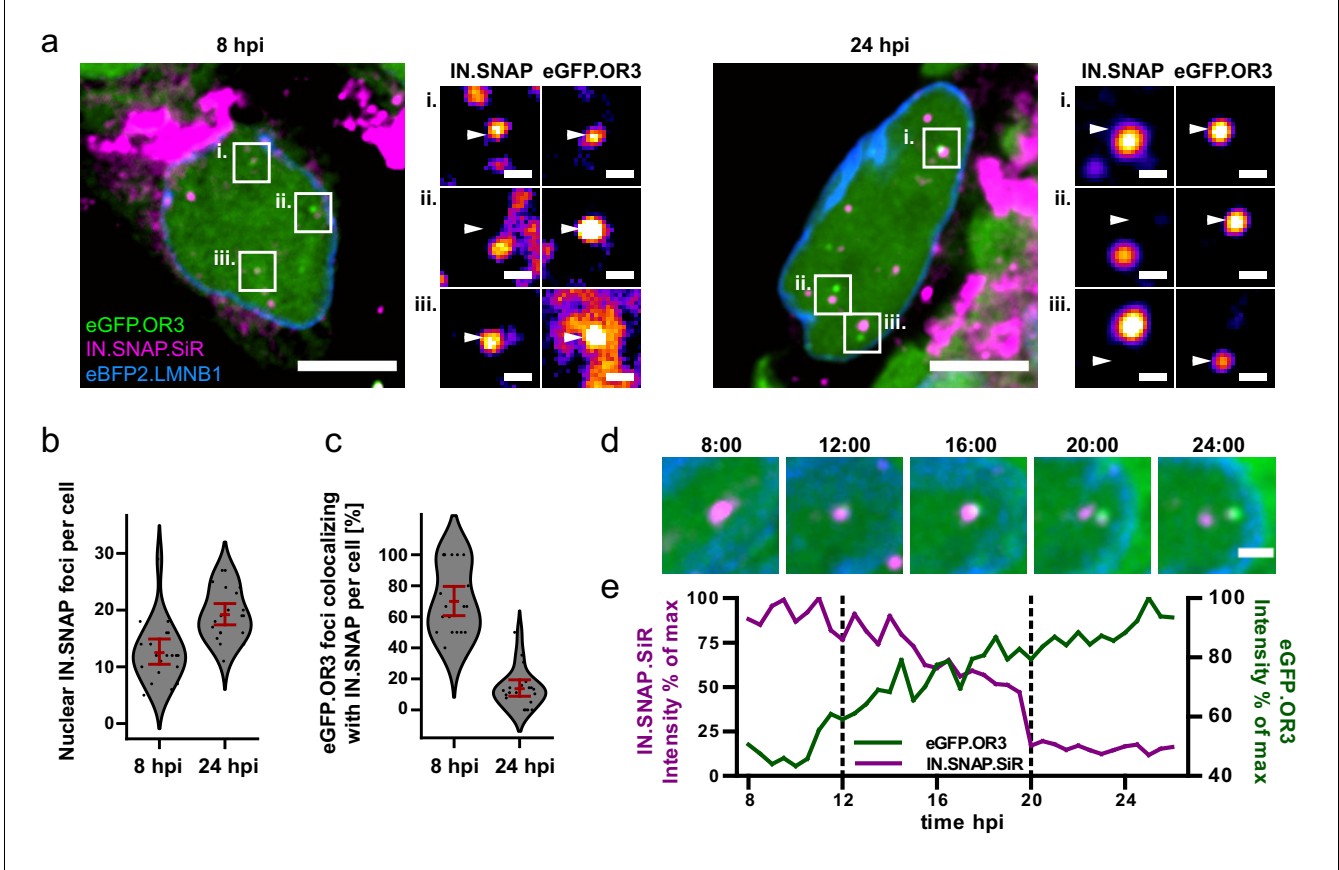

**Figure 3.** Separation of viral cDNA from IN marker within the nucleus. (**a**) TZM-bl eBFP2.LMNB1 eGFP.OR3 cells were infected with VSV-G pseudotyped and IN.SNAP.SiR- labeled NNHIV ANCH particles (30 µU RT/cell), fixed at 8 and 24 h.p.i as indicated and imaged by SDCM. A 1 µm MIP of a representative cell and three enlarged single z slices are shown. Scale bars: 10 µm (MIP) and 1 µm (enlargements). The nuclear background of eGFP. OR3 was subtracted in enlargements for clarity. The figure shows representative images from one of three independent experiments. See *Figure 3— figure supplement 1* for control samples. (**b**) Number of nuclear IN.SNAP punctae/cell detected at the indicated time p.i.; 13 ± 3 punctae/cell (8 h p. i.; ±95 % CI) and 19 ± 2 punctae/cell (24 h p.i.; ±95 % CI). Error bars represent 95 % CI. (**c**) eGFP.OR3 and IN.SNAP co-localization observed over time; 70 ± 11% (8 h p.i.; ±95 % CI) and 14 ± 6% (24 h p.i.; ±95 % CI). Error bars represent 95 % CI. (**d**) Live imaging of IN.SNAP.SiR (magenta) and eGFP.OR3 (green) signal separation within the nucleus. eBFP2.LMNB1 is shown in blue. The figure shows individual frames from *Figure 3—video 1* (4.5 µm MIP) recorded at the indicated times (h:min p.i.). Scale bar: 2 µm; one of three independent experiments. Also see *Figure 3—video 2*. (**e**) Relative intensities of IN.SNAP.SiR and eGFP.OR3 fluorescence detected at the eGFP.OR3 focus in (**d**). The plot comprises data corresponding to the position of the respective IN.SNAP focus recorded before the appearance of eGFP.OR3 (8–11 h p.i.). The area between the dotted lines corresponds to the period of colocalization between the major parts of the IN.SNAP signal and the eGFP.OR3 signal. See *Figure 3—figure supplement 2* for another example. The online version of this article includes the following video, source data, and figure supplement(s) for figure 3:

**Source data 1.** Data corresponds to the number of IN.SNAP.SiR punctae per cell nucleus (*Figure 3b*) and the percent colocalization of eGFP.OR3 punctae with IN.SNAP.SiR per cell nucleus (*Figure 3c*).

**Figure supplement 1.** Different TZM-bl cell lines infected with different particles.

**Figure supplement 2.** Additional example of live imaging separation data.

**Figure 3—video 1.** Live cell imaging of eGFP.OR3 separation from IN.SNAP.
https://elifesciences.org/articles/64776#fig3video1

**Figure 3—video 2.** Live cell imaging of eGFP.OR3 separation from IN.SNAP.
https://elifesciences.org/articles/64776#fig3video2

absolute number of DNA molecules detected by FISH (~80/cell) is likely due to the high density of signals, with some diffraction limited punctae representing more than one reverse transcript. In contrast, only ~4 nuclear eGFP.OR3 clusters were detected per cell (*Figure 1f*). These results clearly showed that the majority of late RT products – including all cytoplasmic products – were not associated with eGFP.OR3.

Progress of reverse transcription was also assessed at the single particle level. For this, we infected cells with NNHIV ANCH carrying fluorescently tagged IN-FP in the presence of EdU followed by fluorescent click labeling of newly synthesized DNA at different time points post infection. Cellular DNA synthesis was inhibited by the DNA polymerase α/δ inhibitor aphidicolin (APC) (*Figure 2d–f*, *Figure 2—figure supplement 1*). Co-localization with EdU was observed for 7% of IN-FP-positive structures in the cytoplasm (95 % CI of mean: 4–9%) and for 36% of IN-FP-positive complexes in the nucleus (95 % CI of mean: 29–43%). Importantly, 81% of nuclear eGFP.OR3 punctae were positive for EdU (95 % CI of mean: 70–93%; *Figure 2d,e*). While some nuclear EdU-positive complexes were positive for both the IN-FP and eGFP.OR3, we also observed HIV-1 cDNA containing complexes that were only positive for either IN-FP or eGFP.OR3 (*Figure 2d*, panels i and ii). The average EdU signal, as a correlate for reverse transcript length, was lower on cytoplasmic than on nuclear HIV-1 complexes and was highest on eGFP.OR3 punctae. When setting the EdU signal for eGFP.OR3 punctae to 100%, the relative EdU signal was significantly reduced to 23% (p<0.0001) on cytoplasmic IN-FP-positive complexes and to 52% (p<0.0001) on nuclear IN-FP-positive complexes (*Figure 2f*). These observations support recent reports that reverse transcription is completed inside the nucleus (*Burdick et al., 2020*; *Dharan et al., 2020*; *Francis et al., 2020*; *Selyutina et al., 2020*) and indicate that the viral cDNA only becomes detectable to eGFP.OR3 when reverse transcription has been completed.

## HIV cDNA separates from IN-fusion proteins in the nucleus of infected cells

The low degree of co-localization between the fluorescent IN fusion protein and eGFP.OR3 on EdU-positive nuclear punctae at 24 h p.i. (*Figure 2d*) prompted us to analyze the relative distribution of both fluorescent proteins in a time resolved manner after NNHIV ANCH infection. At 8 h p.i., 70 ± 11% of nuclear eGFP.OR3 punctae were also positive for IN.SNAP (*Figure 3a–c*). This co-localization was largely lost at 24 h p.i. with only 14 ± 6% of nuclear eGFP.OR3 punctae positive for IN.SNAP (*Figure 3a–c*). Strikingly, IN.SNAP punctae were often observed in close vicinity of eGFP.OR3 punctae at this later time point (*Figure 3a*, right panel), suggesting that they may have separated from a common complex. Similar results were observed for HIV-1 ANCH (*Figure 3—figure supplement 1a*) as well as for an integration competent lentiviral vector containing ANCH (*Figure 3—figure supplement 1b*), and when particles were pseudotyped with HIV-1 Env instead of VSV-G (*Figure 3—figure supplement 1c*). These results showed that the observed phenotype was not dependent on the cytosolic entry pathway or on integration competence. To directly address the possibility of separation of the proviral cDNA from IN.SNAP, we performed live cell imaging of infected cells. We observed gradual loss of the IN.SNAP signal correlating with increased eGFP.OR3 recruitment and eventual separation of eGFP.OR3 punctae and IN.SNAP containing complexes (*Figure 3d,e*, *Figure 3—video 1*). Of note, we occasionally observed consecutive appearance of two individual eGFP.OR3 punctae and their subsequent separation from the same IN.SNAP complex (*Figure 3—figure supplement 2*, *Figure 3—video 2*), suggesting that single diffraction limited IN.SNAP punctae may correspond to multiple cDNA containing subviral HIV-1 complexes, indicating clustering of nuclear complexes as has been recently noted by others (*Francis et al., 2020*; *Rensen et al., 2021*).

## Nuclear IN punctae exhibit a strong signal for CA and CPSF6 and correspond to clustered subviral particles

The observation that the IN.SNAP signal remained as a distinct cluster after separation of the eGFP.OR3-associated viral cDNA suggested that these clusters constitute a stable complex potentially containing other viral and cellular proteins held together by a scaffold. The viral capsid or a capsid-derived structure would be an obvious candidate for such a scaffold. Earlier studies reported a wide range of CA amounts at nuclear complexes (*Chen et al., 2016*; *Chin et al., 2015*; *Francis et al., 2020*; *Hulme et al., 2015*; *Peng et al., 2015*; *Zhou et al., 2011*), while a recent study showed strong nuclear signals for an eGFP.CA fusion protein (*Burdick et al., 2020*) in HeLa-derived cells. We decided to revisit this issue, since we and others had previously observed that CA immunostaining efficiency in the nucleus strongly depended on sample treatment conditions (*Chin et al., 2015*; *Zila et al., 2019*).

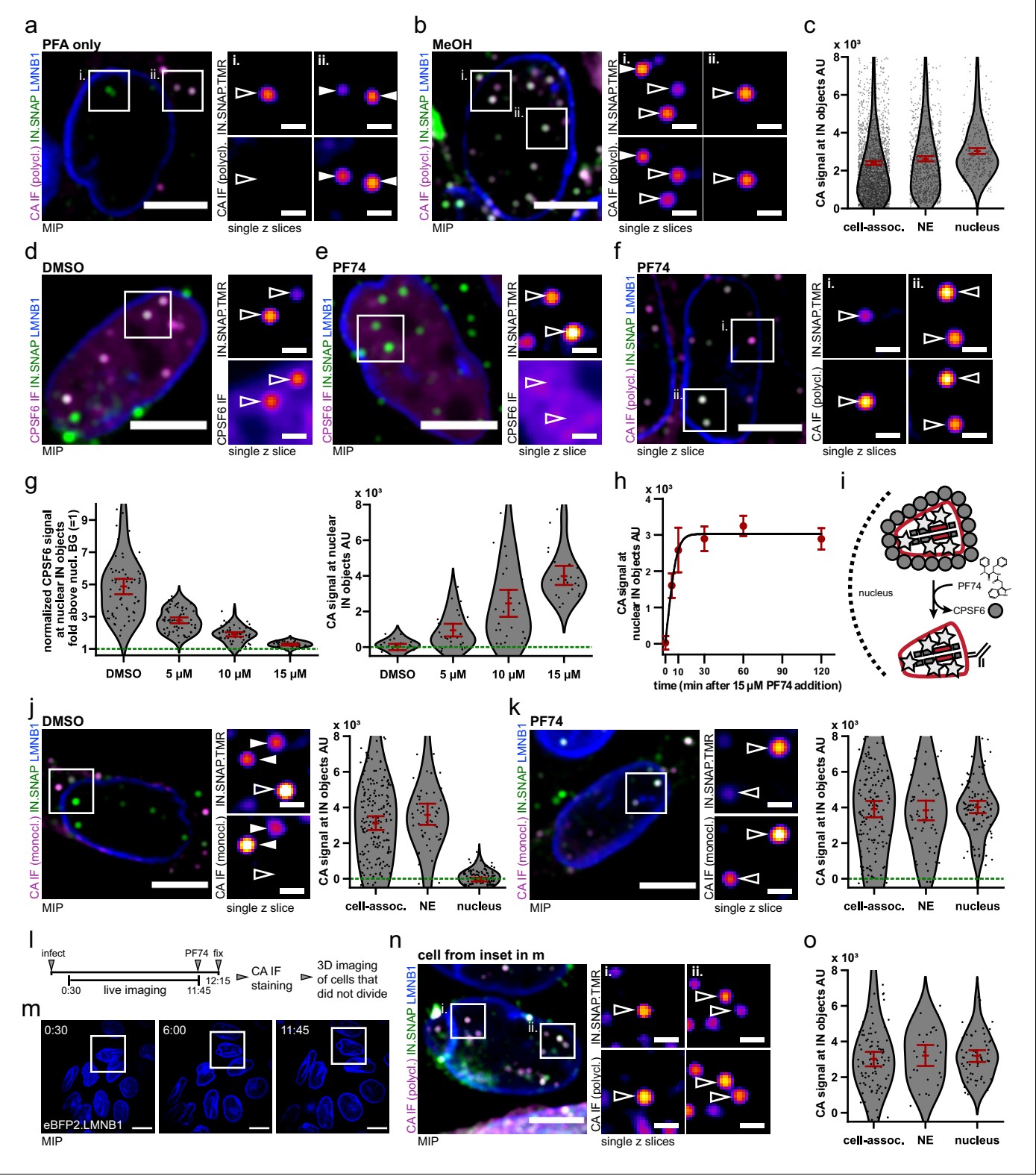

**Figure 4.** HIV-1 CA complexes can enter the nucleus of non-dividing HeLa derived cells through intact NPCs. (**a–c**) IF detection of HIV-1 CA in the nuclei of TZM-bl cells is dependent on extraction with methanol. Cell were infected with VSV-G pseudotyped NNHIV ANCH labeled with IN.SNAP.TMR (30 µU RT/cell), fixed at 24 h p.i. using PFA only (**a**) or extracted using methanol after PFA fixation (**b–c**). Data from one of three independent experiments are shown. (**c**) Quantification of CA intensities at IN.SNAP.TMR objects. Dots represent individual subviral complexes localized at the cytosol/plasma membrane (cell-assoc.), the NE or the nucleus; error bars represent 95 % CI. Cells from five fields of view were analyzed (n = 2610, 932,

*Figure 4 continued on next page*

*Figure 4 continued*

and 286 particles, respectively). Cells had been treated with APC at the time of infection to inhibit division. (**d–i**) PF74-mediated displacement of CPSF6 enables detection of nuclear CA in TZM-bl cells. Infection as in (**a**) for 24 hr followed by 2 hr incubation with DMSO (**d**) or 15 µM PF74 (**e–f**) or the indicated amount of drug or solvent (**g–h**) (see *Figure 4—figure supplement 1a–b*). (**g**) Quantification of nuclear CPSF6 (left panel, n = 215 particles) and nuclear CA (right panel, n = 97 particles) signals at IN.SNAP objects upon treatment with indicated amounts of PF74. CPSF6 signals were normalized to the diffuse nuclear background expression (green line). Dots represent individual subviral complexes localized inside the nucleus; error bars represent 95 % CI. (**h**) Time course of PF74-mediated CA signal appearance. Shown is the mean CA signal at nuclear IN.SNAP objects with error bars indicating 95 % CI fit to a four-parameter logistic equation (as in *Figure 1h*; n = 252 particles) (**i**) Scheme of PF74-mediated CPSF6 displacement. (**j–k**) TZM-bl cells were infected as in (**f**) and treated with DMSO (**j**) or PF74 (**k**) but stained using a monoclonal CA antibody (71-31) (*Figure 4—figure supplement 1c–d*). Graphs show quantification of CA intensities with dots representing individual subviral complexes localized at the cytosol/plasma membrane (cell-assoc.), the NE or the nucleus; error bars represent 95 % CI (n = 200, 56, 113 particles, respectively (**j**); n = 163, 63, 89 particles, respectively (**k**)). (**l–o**) Detection of nuclear CA in non-dividing cells. TZM-bl eBFP2.LMNB1 cells were infected as in (**c**) in the presence of 6 µM APC and live imaging in the eBFP2.LMNB1 channel was initiated recording one frame every 15 min. After 11 hr 45 min cells were treated with 15 µM PF74 for 30 min and then fixed, processed for immunostaining and imaged as in (**f**). Only cells that had not undergone division were analyzed. The figure shows representative images of one of two independent experiments performed in biological duplicates. (**l**) Scheme of experiment (see *Figure 4—figure supplement 1e–f*). (**m**) MIP of exemplary frames from one movie. None of the cells in the FOV displayed nuclear envelope breakdown and mitosis (see *Figure 4—video 1*). Scale bars: 20 µm. (**n**) IF detection of nuclear CA signals in the cell boxed in (**m**). 1 µm MIP and two enlarged z slices are shown. (**o**) Quantification of CA intensities in the subset of non-dividing TZM-bl cells at IN.SNAP.TMR objects. Dots represent individual subviral complexes localized at the cytosol/plasma membrane (cell-assoc.), the NE or the nucleus; error bars represent 95% CI (n = 97, 27, 57 particles, respectively). Scale bars: 10 µm (overviews) and 2 µm (enlargements). Filled arrowheads point at cytoplasmic particles and open arrowheads at nuclear particles.

The online version of this article includes the following video, source data, and figure supplement(s) for figure 4:

**Source data 1.** Data corresponds to the CA IF intensities of IN.SNAP objects at different localizations within the cell (*Figure 4c*), the CPSF6 and CA IF intensities of nuclear IN.SNAP objects treated with different PF74 concentrations for 2 hr (*Figure 4g*), the CA IF intensities of nuclear IN.SNAP objects treated with 15 µM PF74 for different durations (*Figure 4h*), the CA IF intensities of IN.SNAP objects at different localizations within the cell after DMSO or PF74 treatment stained with a monoclonal antibody (71-31) (*Figure 4j–k*) and the CA IF intensities of IN.SNAP objects at different localizations within cell that have not divided (*Figure 4o*).

**Figure supplement 1.** Either methanol extraction or PF74 mediated CPSF6 displacement are sufficient to expose CA epitopes for antibody staining in HeLa-derived cells.

**Figure supplement 1—source data 1.** Data corresponds to the CA IF intensities of IN.SNAP objects at different localizations within the cell (*Figure 4—figure supplement 1a,b*).

**Figure 4—video 1.** Live cell imaging of nuclear integrity in TZM-bl cells.

https://elifesciences.org/articles/64776#fig4video1

---

While PFA fixation alone did not result in CA immunostaining of nuclear complexes (*Figure 4a*), methanol extraction (following PFA fixation) of parallel samples consistently yielded clear CA-specific signals co-localizing with most IN-FP-positive punctae inside the nucleus of TZM-bl cells (*Figure 4b–c* and *Figure 4—figure supplement 1a*). Furthermore, IN-FP-positive punctae were also strongly positive for the host protein cleavage and polyadenylation specificity factor 6 (CPSF6) that binds specifically to the hexameric CA lattice (*Bhattacharya et al., 2014*; *Price et al., 2014*; *Figure 4d*) and is involved in nuclear import (*Bejarano et al., 2019*; *Burdick et al., 2020*; *Chin et al., 2015*), trafficking to nuclear speckles (*Francis et al., 2020*; *Rensen et al., 2021*) and integration-site targeting (*Achuthan et al., 2018*; *Francis et al., 2020*; *Sowd et al., 2016*). We reasoned, that this dense coat of CPSF6 might mask the underlying capsid lattice, precluding antibody detection. This hypothesis was tested by treating infected cells - after the subviral complexes had entered the nucleus – with the small molecule PF74, which acts as a competitive inhibitor of CA-CPSF6 interaction (*Price et al., 2014*) and may thus displace CPSF6 from the subviral complex.

TZM-bl cells infected with NNHIV ANCH for 24 hr were treated with different concentrations of PF74, followed by fixation and IF staining (*Figure 4d–g*). We observed a dose-dependent loss of CPSF6 signal from nuclear HIV-1 subviral complexes upon PF74 treatment, while IN-FP punctae stayed intact (*Figure 4e and g*, left panel). PF74-mediated removal of CPSF6 from the nuclear subviral complexes revealed a strong CA IF signal (*Figure 4f*), with inverse correlation of CPSF6 and CA signal intensities (*Figure 4g*, right panel). CPSF6 displacement occurred rapidly, within 30 min of PF74 treatment (*Figure 4h*). Thus, efficient detection of nuclear CA signals in HeLa-derived cells can be achieved by either general extraction using methanol, or by specific displacement of CPSF6 from subviral complexes. CA immunostaining intensities of particles outside of the nucleus were not affected by PF74 treatment (*Figure 4—figure supplement 1b*), suggesting that the enhanced CA signal on nuclear complexes was not due to structural changes in the capsid lattice induced by the

drug. This conclusion was further supported by using a monoclonal antibody with a defined epitope close to the CPSF6 binding interface (*Gorny and Zolla-Pazner, 1991*). This epitope was readily accessible and not affected by PF74 treatment on cytosolic capsids, whereas CA detection on nuclear subviral complexes by this antibody again required CPSF6 displacement (*Figure 4j,k*). We conclude that CPSF6 shields CA from antibody detection inside the nucleus and CA signal intensity on nuclear complexes is comparable to that on cytoplasmic complexes in HeLa-derived cells.

In the previous experiment (*Figure 4c*), TZM-bl cells had been treated with APC to block cell division, suggesting that nuclear entry of viral complexes in these HeLa-derived cells had occurred through the intact NPC, as observed in terminally differentiated macrophages (*Bejarano et al., 2018*; *Stultz et al., 2017*). In order to prove that nuclear HIV-1 complexes had entered through intact nuclear pores and did not enter during nuclear envelope breakdown in cells that had escaped cell cycle arrest under our conditions (*Figure 4—figure supplement 1e,f*), we observed eBFP2. LMNB1 expressing TZM-bl cells over the whole 12 hr time course of infection by live imaging (*Figure 4l and m*, *Figure 4—video 1*) before PF74 treatment and IF staining. We clearly detected HIV-1 complexes in the nuclei of cells that had not undergone division during the observation period. These nuclear complexes displayed similar CA intensities compared to extranuclear particles (*Figure 4n and o*), supporting the conclusion that subviral complexes comprising most or all of the CA complement of the viral capsid can pass through intact nuclear pores in HeLa-derived cells.

To further analyze the ultrastructure of these nuclear CA-containing complexes, we employed stimulated emission depletion (STED) nanoscopy and CLEM in combination with electron tomography (ET). The mature HIV-1 capsid contains ca. 50% of the total CA content of the intact virion and post-fusion cytoplasmic capsids therefore exhibit a weaker CA signal than complete virions (*Briggs et al., 2004*; *Zila et al., 2019*). Accordingly, CA specific immunofluorescence would be expected to be lower for nuclear complexes compared to cell-associated particles that represent a mixture of post-fusion particles and complete particles at the plasma membrane or endocytosed in the cytosolic region. However, the observed intensity of the CA signal on nuclear HIV-1 complexes was equal to the intensities observed for cell-associated particles (*Figure 4b*). This may be explained by clustering of capsid-derived structures in a single diffraction limited spot. In order to investigate this possibility, we used STED nanoscopy to resolve individual subviral structures with a resolution of <50 nm. Multiple individual CA signals in close vicinity to each other could be resolved within the area of a single focus detected in confocal mode (*Figure 5a and b*). For a more detailed analysis of the associated structures, we performed CLEM-ET as described in the following section. IN.SNAP-positive and eGFP.OR3-negative nuclear punctae detected by fluorescence microscopy could be correlated with single or multiple electron-dense cone-shaped structures, whose shapes and dimensions closely resembled mature HIV-1 capsids (*Figure 5c*, *Figure 5—video 1*). STED nanoscopy of nuclear SNAP.OR3-positive punctae corresponding to viral cDNA identified elongated structures (*Figure 5d*); in this case, only a single object was resolved by STED nanoscopy at each diffraction-limited position.

## Ultrastructure analysis reveals conical and elongated structures in the position of IN punctae and OR3 punctae, respectively

The findings described above suggest that apparently intact conical HIV-1 capsids can access the nucleoplasm, where reverse transcription is completed followed by separation of the viral cDNA from the bulk of viral proteins including CA. To determine the ultrastructure of the observed subviral complexes, we performed CLEM-ET analysis. For this, we employed a TZM-bl mScarlet.OR3 cell line (*Figure 3—figure supplement 1d*), because the mScarlet fluorescence signal was best retained upon plastic embedding. Cells were infected with VSV-G pseudotyped NNHIV ANCH carrying IN.SNAP.SiR, thus allowing direct high-pressure freezing of the sample without pre-fixation. Cells were vitrified at 24 h p.i. and thin sections were prepared after freeze-substitution and plastic embedding. Samples retained fluorescence for mScarlet.OR3 and IN.SNAP.SiR. Multi-channel fluorescent Tetraspeck markers were used for correlation and we identified positions corresponding to mScarlet.OR3 and IN.SNAP.SiR signals, respectively (*Figure 6a–c*). These positions were imaged using electron tomography (*Figure 6d–i*). A total of 21 individual structures were identified and visualized by CLEM-ET covering the volume of six individual nuclei (*Figure 6—figure supplement 1*). The majority of structures (20/21; 95%) was found in clusters of two or more structures. At positions correlating to IN-positive punctae lacking mScarlet.OR3 (18/21; 86%), we detected 10 cone-shaped structures

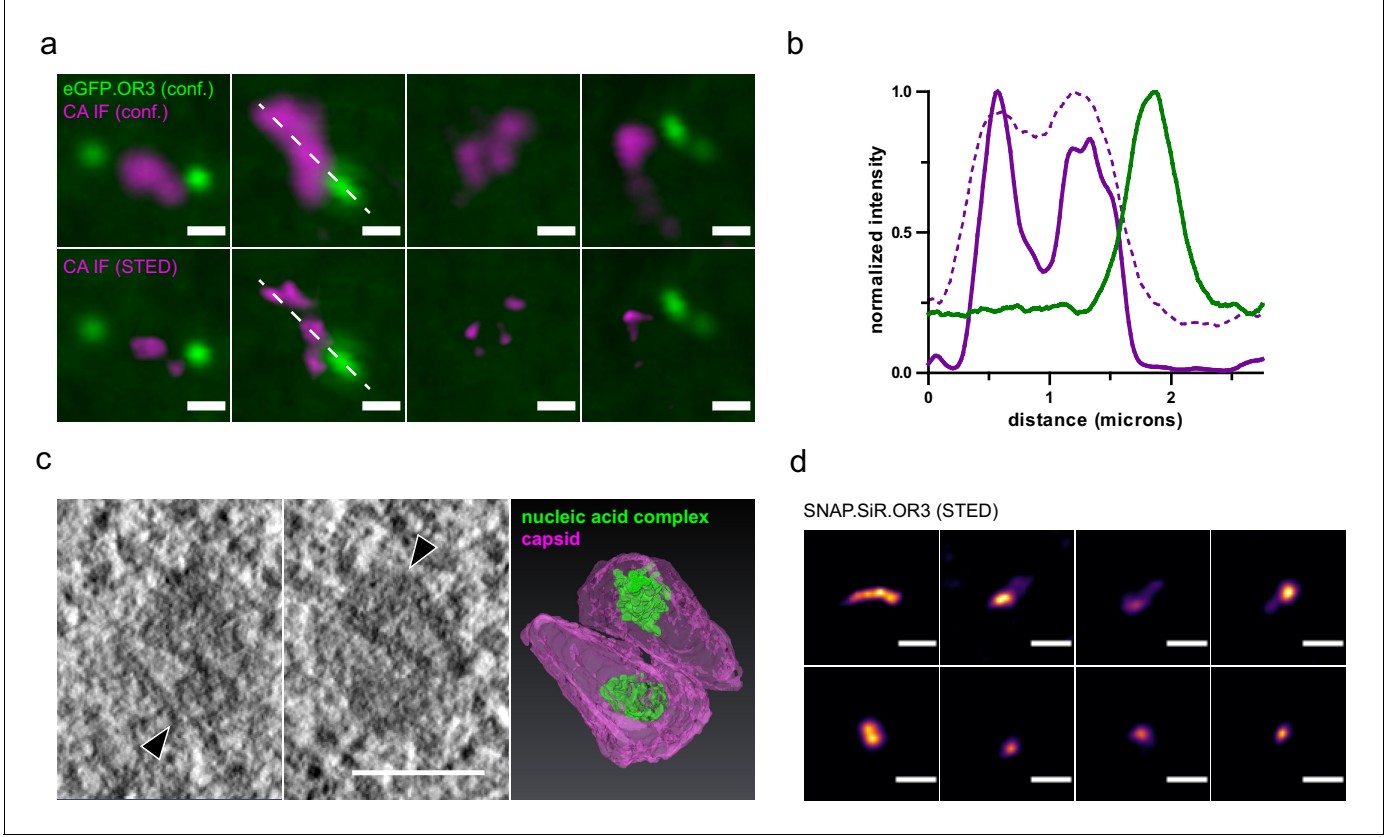

**Figure 5.** HIV-1 capsids cluster within the nucleus of HeLa derived cells. (**a**) STED nanoscopy of CA accumulation at diffraction limited spots. Shown are four examples of nuclear CA and OR3 signals. TZM-bl eBFP2.LMNB1 and eGFP.OR3 cells were infected using VSV-G pseudotyped NNHIV ANCH labeled with IN.SNAP.TMR (30 μU RT/cell) and fixed at 24 h p.i.; bottom panel shows STED microscopy of CA signals. Scale bars: 500 nm (**b**) Intensity profiles measured along the dashed white line in (**a**) normalized to the respective maximal value. Magenta, CA intensities in STED mode; dashed magenta, CA intensities in confocal mode; green, eGFP.OR3 intensities in confocal mode. (**c**) Electron tomography of NNHIV capsids detected in the nucleus. eGFP.OR3 expressing TZM-bl cells were infected with VSV-G pseudotyped NNHIV ANCH labeled with IN.SNAP.SiR (30 μU RT/cell) and high pressure frozen at 24 h p.i. Left and middle panels show slices through a tomographic reconstruction at the position within the nucleus correlated to an IN.SNAP spot (negative for eGFP.OR3). Arrowheads point to two closely associated cone-shaped structures where the wide end of one cone is oriented toward the narrow end of the other cone. Scale bar: 100 nm. The right panel shows the segmented and isosurface rendered structure of these cones. Magenta, capsid; green, nucleic acid containing replication complex. See *Figure 5—video 1*. (**d**) STED nanoscopy of SNAP.OR3 expressing TZM-bl cells infected with VSV-G pseudotyped NNHIV ANCH (30 μU RT/cell) and fixed at 24 h p.i. Shown is a selection of eight nuclear SNAP.OR3 signals stained with O6-benzylguanine (BG)-SiR for 30 min prior fixation and analyzed by STED nanoscopy. Note that nuclear background of SNAP.OR3 has been subtracted for clarity. Scale bars: 300 nm.

The online version of this article includes the following video for figure 5:

**Figure 5—video 1.** CLEM-ET analysis, segmentation, and rendering of nuclear capsid cone clustering in an infected HeLa-derived cell.

https://elifesciences.org/articles/64776#fig5video1

resembling intact HIV-1 capsids (*Figures 5c* and *6g* ii. and 6i.) and eight structures with less defined morphology consistent with deformed tubular structures or remnants of capsids (*Figure 6g* ii and 6h ii; *Figure 6—video 1*). Dense material most likely corresponding to the viral nucleoprotein complex was visible inside of most conical structures (*Figures 5c* and *6g* ii. and 6i top left panel, black arrowheads), whereas tubular and capsid remnant-like structures mostly lacked interior densities (*Figure 6g* ii., 6h ii. and 6i bottom left panel, open white arrowheads).

In contrast, electron tomograms that correlated to positions of mScarlet.OR3 signals (lacking IN. SNAP.SiR signal; n = 7) showed no defined conical or capsid remnant-like structures. Instead, elongated dense objects of ~100–300 nm in length were observed (*Figure 6g* i. and 6h i.), in line with the findings from STED nanoscopy (compare *Figure 5d*). These structures consisted of linked globular densities with a diameter of ~30 nm resembling the appearance of chromatin (*Figure 6h* i.).

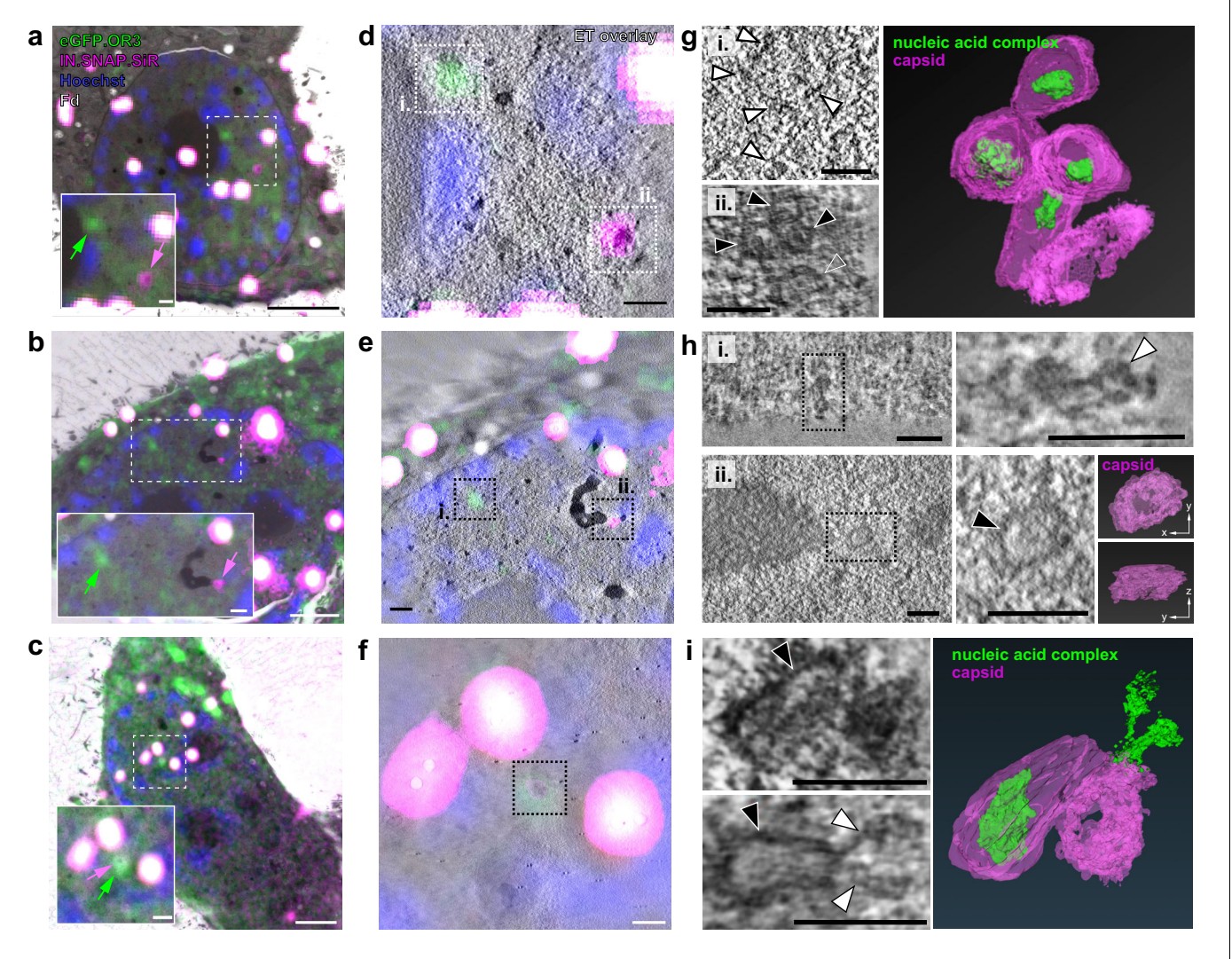

**Figure 6.** CLEM-ET analysis of IN and OR3 punctae inside the nucleus of infected HeLa-derived cells. TZM-bl cells expressing mScarlet.OR3 were infected with VSV-G pseudotyped and IN.SNAP.SiR-labeled NNHIV ANCH (30 µUnits RT/cell). At 24 h p.i., cells were cryo-immobilized by high pressure freezing, freeze substituted and further processed for CLEM and ET. (a–c) CLEM overlays (with enlargements) of EM sections of cells expressing mScarlet.OR3 (green), infected with NNHIV ANCH IN.SNAP.SiR (magenta), post-stained with Hoechst (blue) and decorated with multi-fluorescent fiducials (Fd; white) for correlation. Positions of intranuclear spots positive for mScarlet.OR3 (green arrows) and IN.SNAP.SiR (magenta arrows) are indicated. Enlargements of area in dashed boxes is shown at the lower left of each panel. (d–f) CLEM-ET overlay of regions enlarged in (a–c). (g–i) Computational slices from tomographic reconstructions at the correlated positions boxed in (d–f). (g) Top panel (i.), white arrowheads point to a filamentous structure corresponding to an mScarlet.OR3 (and IN.SNAP negative) spot boxed in (d; i. - green arrow in (a)). Bottom panel (ii.), black arrowheads indicate capsid-reminiscent structures and the open arrowhead indicates a remnant-like tubular structure lacking internal density correlating to the IN.SNAP.SiR spot boxed in (d; ii. - magenta arrow in (a)). The right panel shows the segmented and isosurface rendered structures shown in (d; ii.). See *Figure 6—video 1*. (h) Top panels show a chromatin-like density (white arrowhead), consisting of apparently linked globular structures, correlating to the mScarlet.OR3-positive and IN.SNAP.SiR-negative spot boxed in (e; i.). Lower panels show the morphology of an empty open structure (black arrowhead) correlating to the IN.SNAP.SiR positive, mScarlet.OR3 negative spot boxed in (e; ii.). The right panels show the segmented and isosurface rendered structure shown in (e; ii.). (i) Morphology of structures clustering at the position indicated by co-localizing mScarlet.OR3 and IN. SNAP.SiR spots boxed in (f). Top left, black arrowhead indicates an apparently intact capsid with density inside the cone. Bottom left, the black arrowhead indicates an apparently empty cone-like structure. Note an elongated density (white arrowhead) protruding from the narrow end of the cone. The right panel shows the segmented and isosurface rendered structures shown on the left. See *Figure 6—video 2*. Scale bars: 2.5 µm for overviews (a–c), 500 nm for enlargements (a–c), 250 nm (d–f), and 100 nm (g–i). See *Figure 6—figure supplement 1* for a gallery of all visualized capsid structures.

The online version of this article includes the following video and figure supplement(s) for figure 6:

**Figure supplement 1.** Overview of viral capsid structures captured by CLEM-ET inside the nucleus of infected HeLa-derived cells.

*Figure 6 continued on next page*

*Figure 6 continued*
**Figure 6—video 1.** CLEM-ET analysis, segmentation and rendering of IN-positive (and OR3-negative) structures in an infected HeLa-derived cell.
https://elifesciences.org/articles/64776#fig6video1
**Figure 6—video 2.** CLEM-ET analysis, segmentation, and rendering of IN- and OR3-positive structures in an infected HeLa-derived cell.
https://elifesciences.org/articles/64776#fig6video2

One of the visualized objects correlated to both IN.SNAP.SiR and mScarlet.OR3. The corresponding electron tomogram revealed a dense cluster of three capsid-related structures (*Figure 6i*). One of these structures lacked interior density (*Figure 6i*, bottom left panel, open arrowhead) and appeared to be connected with an adjacent elongated density (filled white arrowheads) that seemed to protrude from the narrow end of the cone (*Figure 6i* right panel; *Figure 6—video 2*). Taken together with the observations from live cell imaging, we speculate that this structure might represent a subviral complex captured in the process of capsid uncoating and genome release.

## Nuclear CA and segregation of HIV-1 cDNA from the bulk of viral proteins are also observed in HIV-1 infected SupT1 cells, primary CD4[+] T cells and monocyte-derived macrophages

In order to validate our findings in more relevant cell types, we adapted the system to the T cell line SupT1 and to primary CD4[+] T cells and primary monocyte-derived macrophages (MDM). Of note, nuclear CA immunofluorescence signals have been detected previously in MDM (*Bejarano et al., 2019*; *Francis et al., 2020*), but not or only weakly in T cell lines (*Zila et al., 2019*) or primary T cells. Infection of an eGFP.OR3 expressing SupT1 cell line (*Figure 7—figure supplement 1*) or of primary activated CD4[+] T cells transduced to express eGFP.OR3 (*Figure 7a*) with NNHIV ANCH showed nuclear OR3 punctae and separation of IN-FP and OR3 punctae as observed for TZM-bl cells.

In accordance with previous observations (*Zila et al., 2019*), no or weak CA signals were detected co-localizing with IN.SNAP punctae in T cells (*Figure 7a*), even upon methanol extraction. However, similarly to observations in HeLa-based cells (*Figure 4d*), strong signals for CPSF6 were detected at positions of IN-FP (*Figure 7b* top panel). Displacing this CPSF6 coat from the nuclear subviral particles using PF74 (*Figure 7b–c*) as described above, strikingly revealed a strong CA IF signal (*Figure 7d–f*). Of note, detection of nuclear CA in T cells depended on a combination of PF74 treatment and methanol extraction of specimens (*Figure 7—figure supplement 2*), while either treatment alone was sufficient to expose CA in HeLa-derived cells (*Figure 4—figure supplement 1a–b*). In agreement with results from HeLa-based cells, no difference was observed for CA IF signals on extracellular particles and on cytoplasmic or nuclear envelope-associated subviral complexes upon PF74 treatment, while the nuclear CA signal became strongly enhanced (*Figure 7e–f*). Thus, the failure to detect nuclear CA by IF in CD4[+] T cells is due to shielding of epitopes by the accumulation of CPSF6 rather than to CA being lost upon nuclear entry in these cells, in accordance with our recent cryo-ET analyses in SupT1 cells (*Zila et al., 2021*).

Compared to extracellular virions, subviral particles in the cytosol and at the nuclear envelope displayed a reduced CA signal due to loss of free CA molecules from the post-fusion complex. This phenotype was most notable for nuclear envelope-associated complexes at later time points whose CA signal intensity corresponded to ~50% of that of complete particles (*Figure 7f*). Of note, nuclear subviral complexes showed a higher mean CA signal compared to complete virions, consistent with nuclear clustering of CA containing complexes (*Francis et al., 2020*; *Rensen et al., 2021*) also in this cell type; this was most evident at earlier time points (*Figure 7e–f*).

After PF74 treatment we could also observe some nuclear eGFP.OR3 punctae representing HIV-1 cDNA that were associated with IN-FP and strongly CA positive (*Figure 7g*). Upon separation from eGFP.OR3 punctae, CPSF6 clusters remained associated with the IN-FP, confirming that the viral cDNA separates from an IN-FP/CA/CPSF6-positive nuclear structure in the nucleus of infected primary CD4[+] T cells as well (*Figure 7h*).

Finally, we adapted the ANCHOR system to primary MDM by transducing these cells with an eGFP.OR3 expressing lentiviral vector. Three days post transduction, MDM were infected with VSV-G pseudotyped NNHIV ANCH containing IN.mScarlet for 24 hr. A similar phenotype as shown

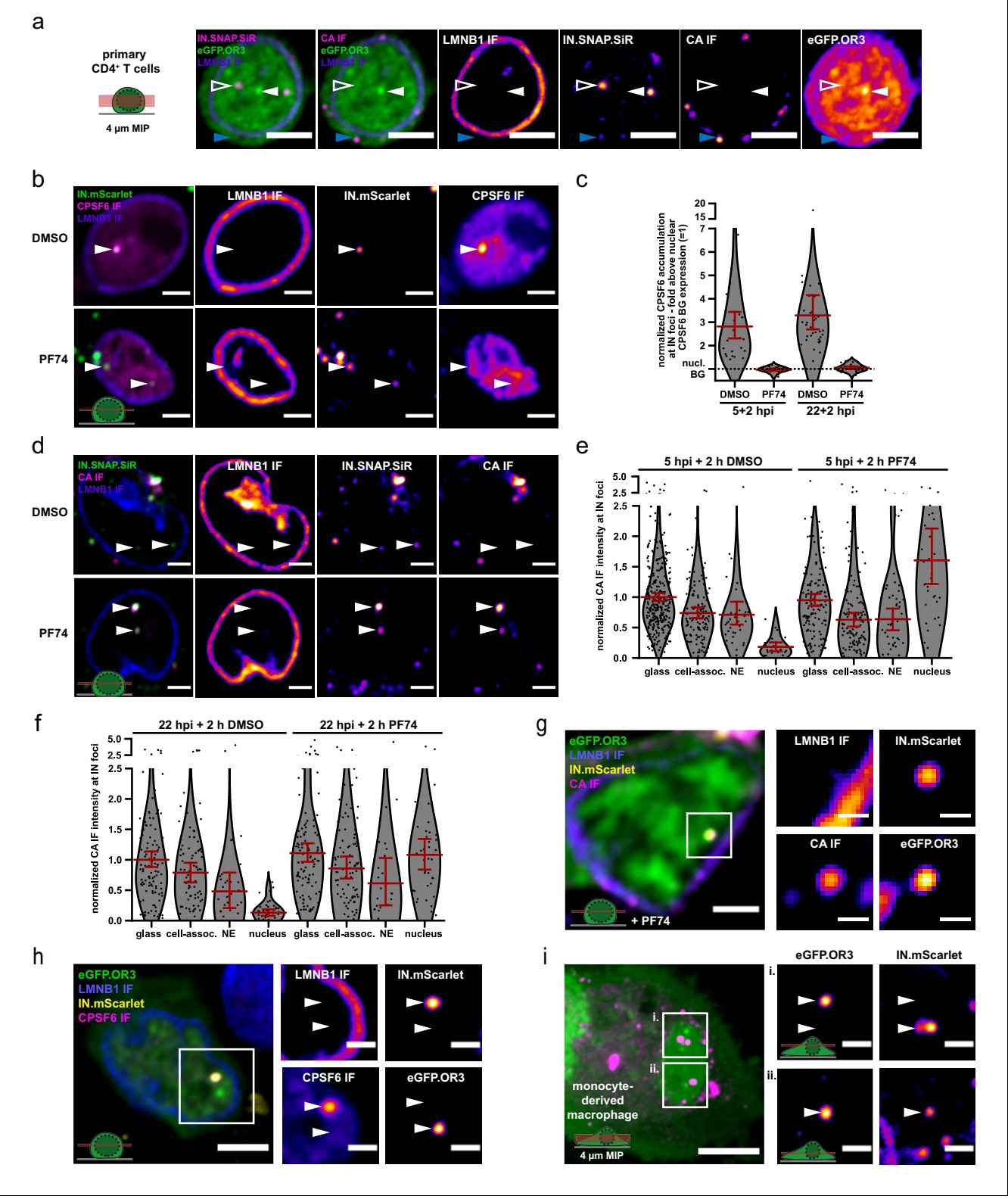

**Figure 7.** HIV DNA separates from IN/CA/CPSF6-positive structures in primary CD4[+] T cells and MDM. (a) Activated CD4[+] T cells were transduced with a lentiviral vector expressing eGFP.OR3. After 48 hr, cells were infected using VSV-G pseudotyped NNHIV ANCH labeled with IN.SNAP.SiR (30 μU RT/cell) for 24 hr before fixation and methanol extraction (as in *Figure 4a–b*). Arrowheads indicate the positions of a nuclear (left, open white) and cytoplasmic (bottom, blue) IN.SNAP particle as well as a nuclear eGFP.OR3 focus (right, filled white). A representative image of one of three

*Figure 7 continued on next page*

Figure 7 continued

independent experiments is shown. Scale bars: 5 µm. See *Figure 7—figure supplement 1* for SupT1 cells. (**b–f**) Addition of PF74 (15 µM) after nuclear import enables immuno-detection of strong CA signals in CD4+ T cells. Activated primary CD4+ T cells were infected using VSV-G pseudotyped NNHIV ANCH IN.mScarlet or IN.SNAP.SiR (30 µU RT/cell). PF74 or DMSO was added after 5 or 22 hr for another 2 hr prior to fixation and methanol extraction. Shown is one of two independent experiments performed with cells from three different blood donors. (**b**) PF74 displaces CPSF6 from IN spots. Shown are single z slices from cells fixed at 24 h p.i. and immunostained for CPSF6. Scale bars: 3 µm (**c**) Quantification of CPSF6 intensity at IN spots. IN objects were segmented in 3D data sets. The CPSF6 mean intensity of these volumes was quantified as described in Materials and methods. Dots represent single subviral complexes and error bars represent 95 % CI. (**d**) PF74 enables CA IF detection at IN spots. Shown are single z slices from cells fixed at 24 h p.i. and immunostained for HIV-1 CA (see *Figure 7—figure supplement 2*). Scale bars: 3 µm. (**e–f**) Quantification of CA intensities at IN spots at 7 hr (**e**) and 24 hr (**f**) p.i. IN-positive objects were segmented in 3D. The CA mean intensity of these volumes was quantified and normalized to the CA intensity of IN-positive objects located on glass in DMSO-treated samples. Error bars represent 95 % CI. P values of differences between DMSO and PF74 treatments (two-tailed Student's t-test): glass = 1.000 (**e**, not significant (ns)), 1.000 (**f**, ns); cell-assoc.=0.2684 (**e**, ns), 0.9427 (**f**, ns); NE = 0.7884 (**e**, ns), 0.7514 (**f**, ns); nucleus = <0.0001 (**e**, significant),<0.0001 (**f**, significant). (**g**) CA-positive structure colocalizing with IN and vDNA markers inside the nucleus of CD4+ T cells. PF74 was added for 2 hr prior to fixation at 24 h p.i. Nuclear background of OR3 was subtracted in enlargements for clarity (in **g–i**). Shown is a single z slice, scale bars: 5 µm (overview) and 1 µm (enlargement). (**h**) CPSF6 remains colocalized with the IN signal. Shown is a single z slice, scale bars: 5 µm (overview) and 2 µm (enlargement). (**i**) MDM were transduced using a Vpx containing eGFP.OR3 expressing lentiviral vector. After 72 h cells were infected using VSV-G pseudotyped NNHIV IN.mScarlet (120 µU RT/cell), fixed and imaged at 24 h p.i. Shown is a 4 µm maximum intensity projection. Scale bar: 5 µm. Enlargements represent a single z slice; scale bars: 2 µm.

The online version of this article includes the following source data and figure supplement(s) for figure 7:

**Source data 1.** Data corresponds to the normalized (to the nuclear expression) CPSF6 intensities of IN.SNAP.SiR objects (*Figure 7c*), and to the normalized (to the mean intensity of glass-bound particles of the DMSO sample) CA intensities of IN.SNAP.SiR objects at different locations at 7 h p.i. (*Figure 7e*) and at 24 h p.i. (*Figure 7f*).

**Figure supplement 1.** ANCHOR system adopted to the T cell line SupT1.

**Figure supplement 1—source data 1.** Data corresponds to the number of eGFP.OR3 objects and percent of EdU-positive eGFP.OR3 objects per cell nucleus (*Figure 7—figure supplement 1b*).

**Figure supplement 2.** Efficient detection of nuclear CA in T cells depends on the combination of PF74 treatment with methanol extraction.

**Figure supplement 2—source data 1.** Data corresponds to the CA IF intensities of nuclear IN.mScarlet objects with different treatments in SupT1 cells (*Figure 7—figure supplement 2a*) and primary CD4+ T cells (*Figure 7—figure supplement 1b*).

above for TZM-bl cells and primary CD4+ T cells was observed in MDM: the majority of eGFP.O3 punctae was detected separated from, but often in close vicinity of IN.mScarlet punctae (*Figure 7i*).

## Discussion

In this study, we adapted a single molecule labeling method to study the dynamics of HIV-1 cDNA in living cells. Using this system, we showed that the HIV-1 ANCH dsDNA recognizing OR3 marker is only recruited to the viral cDNA inside the nucleus, while no OR3 punctae were observed associated with viral structures in the cytosol despite the presence of abundant reverse transcription products. Both, integrated and unintegrated HIV-1 cDNA were detected by OR3 in the nucleus. Metabolic labeling of nascent DNA revealed that nuclear eGFP.OR3 punctae contained significantly higher DNA amounts compared to cytoplasmic or nuclear subviral complexes lacking the eGFP.OR3 signal. Together with recent reports employing indirect RT inhibitor time-of-addition assays, which showed that reverse transcription remains sensitive to inhibition until after nuclear import (*Burdick et al., 2020*; *Dharan et al., 2020*; *Francis et al., 2020*), and an elegant experiment showing that positive and negative strand-specific HIV DNA hybridisation probes only co-localize inside the nucleus (*Dharan et al., 2020*), our data support completion of HIV-1 reverse transcription in the nucleus. Furthermore, HIV-1 cDNA separated from an IN fusion protein (IN-FP), often used as a marker for the HIV-1 replication complex, inside the nucleus; IN fusions are thus not suitable for tracking HIV-1 cDNA in the nucleus. We expect that unfused IN, also present in the replication complex, will remain – at least partially – with the cDNA to mediate chromosomal integration and thus separates from the bulk of IN-FP. Nuclear IN-FP punctae were strongly positive for the viral CA protein and CA, CPSF6 and IN-FP signals stayed together after separation from the viral cDNA.

OR3 recruitment to the HIV-1 cDNA requires the dsDNA to be accessible to the 66 kDa eGFP. OR3 fusion protein, which depends on loss of integrity of the capsid shell. CA signal intensity on nuclear IN-FP-positive and eGFP.OR3-negative structures was equal to or higher than observed for cytoplasmic complexes, suggesting that the bulk of CA stays associated with the viral replication

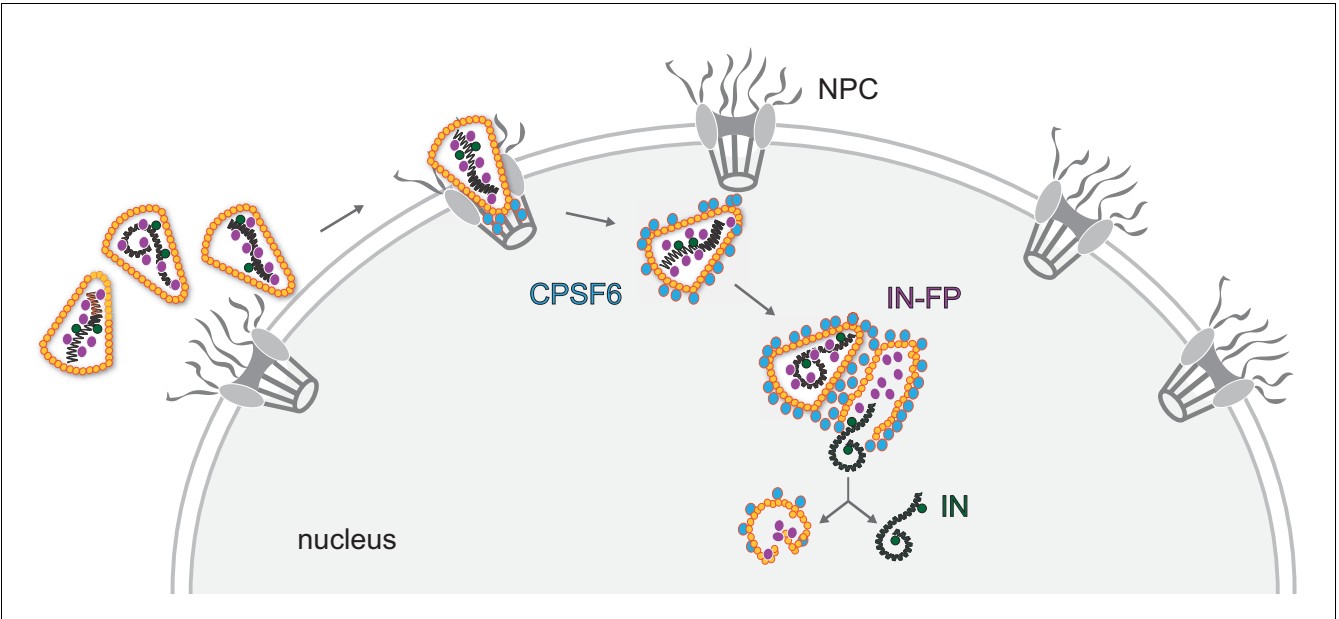

**Figure 8.** Model of HIV-1 nuclear entry and uncoating. Apparently intact HIV capsids are imported into the nucleus through nuclear pore complexes retaining their cone-shaped morphology. CPSF6 releases the cores from the NPC and clusters on nuclear capsids. Multiple capsids accumulate at certain positions within the nucleus, and plus strand synthesis of the viral double-stranded cDNA is completed in the nucleus. Physical disruption of the capsid releases the completed cDNA into the nucleoplasm, where it becomes integrated into the host cell genome in the vicinity of the uncoating site. Empty remnants of the broken capsid, associated with incorporated viral proteins that are not part of the cDNA complex, remain as distinct structures in the nucleus for prolonged times after uncoating.

complex and the viral DNA remains encased inside a closed capsid or capsid-like structure until after nuclear entry. Using CLEM-tomography of IN-positive and eGFP.OR3-negative nuclear complexes, we observed morphologically intact cone-shaped structures with internal density representing the nucleoprotein complex, which closely resembled HIV-1 capsids inside the cytosol or in complete virions. This finding is consistent with our recent study showing that the nuclear pore channel is sufficiently large to accommodate the HIV-1 core and apparently intact cone-shaped HIV-1 capsids can enter the nucleus through intact nuclear pores (*Zila et al., 2021*). Taken together, these results indicate that reverse transcription initiates in the cytoplasm inside a complete or largely complete capsid, and this capsid-encased complex traffics into the nucleus, where reverse transcription is completed; subsequently, it must be uncoated for integration to occur. Formation of eGFP.OR3 punctae requires both, completion of reverse transcription and – at least partial – uncoating, and these two events may conceivably occur in a coordinated manner.

IN-FP (and CA) positive nuclear complexes at later time points were often observed in close vicinity but clearly separated from eGFP.OR3 punctae, and live cell microscopy confirmed separation of the two markers from a single focus over time. Both markers retained their focal appearance and could thus be analyzed by CLEM. Electron tomography of late IN-FP-positive complexes revealed electron-dense structures that resembled broken HIV-1 capsids or capsid-like remnants lacking the density of the nucleoprotein complex. In contrast, eGFP.OR3-positive and IN-negative nuclear subviral structures never exhibited an electron-dense lining resembling the capsid shell, and these complexes were always negative for CA by immunofluorescence. These results indicate that viral cDNA associated with some replication proteins emanates from the broken capsid, which retains the bulk of CA and capsid-associated proteins. Uncoating therefore does not appear to occur by cooperative disassembly of the CA lattice, but by physically breaking the capsid shell and loss of irregular capsid segments. We often observed clustering of capsids or capsid-remnants inside the nucleus indicating

preferred trafficking routes of subviral complexes, as described by others (*Francis et al., 2020*; *Rensen et al., 2021*).

The broken capsid-remnant structures inside the nucleus of infected cells closely resembled ruptured HIV-1 cores observed in a recent study analyzing HIV-1 cDNA formation and integration in an in vitro system using purified virions (*Christensen et al., 2020*). These authors reported partially broken capsid shells with irregular defects at time points when endogenous cDNA formation was largely completed; they also detected polynucleotide loops emanating from the holes in the capsid lattice. Theoretical models and AFM-studies had suggested that the volume of double-stranded HIV-1 b-DNA cannot be accommodated inside the intact capsid and that the resulting pressure might mechanically trigger uncoating (*Rankovic et al., 2017*; *Rouzina and Bruinsma, 2014*). Taken together, these results suggest that the growing dsDNA inside the viral capsid in the nucleus may eventually lead to local rupture of the capsid lattice, concomitantly allowing completion of reverse transcription and triggering uncoating of the proviral DNA. It must be kept in mind, however, that lentiviral vectors with much shorter length of the vector RNA efficiently transduce cells; On the other hand, the effect might be offset by discontinuities present within the growing dsDNA chain in case of the full length genome (*Miller et al., 1995*). Nuclear import is not required for completion of cDNA synthesis and loss of capsid integrity since similar structures were detected in the in vitro system (*Christensen et al., 2020*). The observation that the viral cDNA was not fully released from the viral core in vitro suggests, however, that the nuclear environment may play a role in this process. The described pathway appears to be conserved in HeLa reporter cells and primary HIV-1-sensitive CD4[+] T-cells and MDM: separation of IN/CA complexes from the OR3-positive cDNA inside the nucleus of infected cells was observed in all cell types, and the IN-positive subviral complexes exhibited a strong CA signal in all cases. Previous failure to detect CA on nuclear complexes in T cells has been due to epitope masking by the cellular CPSF6 protein and the current results thus indicate a common pathway for early HIV-1 replication in different cell types including primary target cells of HIV-1 infection.

The efficiency of CA immunostaining in the nucleus appears to be partially dependent on the cell type. In MDMs (*Bejarano et al., 2018*; *Stultz et al., 2017*; *Zila et al., 2019*) and THP-1 cells (*Rensen et al., 2021*), nuclear CA signals were detected by immunofluorescence upon copper-click-mediated cellular extraction without CPSF6-removal. In HeLa-derived cells, immunostaining of nuclear CA could be recovered by either methanol extraction or PF74-mediated CPSF6 displacement, whereas a combination of methanol extraction and PF74 treatment was necessary in T-cells. This difference might be explained by differences in nuclear architecture, the architecture of the CPSF6 cluster or/and the presence of other cell-type-specific proteins in the cluster and warrants further investigation. Nevertheless, strong CA signals resembling the intensity on cytoplasmic subviral HIV-1 complexes were detected in all cell types indicating a common pathway for nuclear entry of capsid-containing HIV-1 replication structures.

STED and CLEM analysis revealed elongated structures with regularly spaced globular densities at the position of eGFP.OR3-positive punctae that had separated from the IN fusion protein and CA. These structures resembled chromatinized DNA (*Miron et al., 2020*), in line with biochemical evidence that HIV-1 cDNA is rapidly chromatinized when it becomes accessible to the nucleoplasm (*Geis and Goff, 2019*). Detection of a cone-shaped structure lacking the electron-dense internal nucleoprotein signal and directly associated with an elongated chromatin-like structure at a position that was positive for both IN-FP and eGFP.OR3 may have captured a subviral complex in the process of uncoating. We suggest that chromatinization of HIV-1 cDNA emerging from the broken capsid may facilitate complete uncoating of the genome, which could explain why viral cDNA remained largely associated with the capsid structure in the in vitro system.

In conclusion, our results indicate that complete or largely complete HIV-1 capsids enter the nucleus of infected cells, where reverse transcription is completed and the viral cDNA genome is released by physical disruption rather than by cooperative disassembly of the capsid lattice (*Figure 8*). The viral capsid thus plays an active role in the entire early phase of HIV-1 replication up to chromosomal integration and appears to be important for cytoplasmic trafficking, reverse transcription, shielding of viral nucleic acid from the innate immune system, nuclear entry, and intranuclear trafficking. The cone-shaped HIV-1 capsid with its fullerene geometry thus is the key orchestrator of early HIV-1 replication.

# Materials and methods

## Key resources table

| Reagent type (species) or resource | Designation | Source or reference | Identifiers | Additional information |
|---|---|---|---|---|
| Antibody | Anti-HIV-1 CA (rabbit polyclonal) | In-house; *Bejarano et al., 2019*; doi:10.7554/eLife.41800 | - | IF (1:1000) |
| Antibody | Anti-HIV-1 CA 71–31 (human monoclonal) | NIH ARP; *Gorny et al., 1989*; doi:10.1073/pnas.86.5.1624 | NIH:ARP-530 | IF (1:500) |
| Antibody | Anti-hCPSF6 (rabbit polyclonal) | Atlas Antibodies | Cat#:HPA039973 RRID:AB_10795242 | IF (1:500) |
| Antibody | Anti-hLamin A/C (mouse monoclonal) | Santa Cruz Biotech. | Cat#:sc-7292 RRID:AB_627875 | IF (1:100); works for HeLa cells and MDM |
| Antibody | Anti-hLaminB1 (mouse monoclonal) | Santa Cruz Biotech. | Cat#:sc-365962 | IF (1:200); works for SupT1 and primary CD4$^+$ T cells |
| Antibody | Alexa Fluor 405, 488, 568 and 647 secondary antibodies (goat polyclonal) | Thermo Fisher Scientific | - | IF (1:1000) |
| Antibody | Anti-rabbit IgG Atto 594 (goat polyclonal) | Sigma Aldrich | Cat#:77671 | IF (1:500); STED |
| Cell line (*Homo sapiens*) | Hela TZM-bl | *Wei et al., 2002*; doi:10.1128/AAC.46.6.1896-1905.2002 | RRID:CVCL_B478 | - |
| Cell line (*H. sapiens*) | Embryonic kidney 293 T cells (HEK293T) | *Pear et al., 1993*; doi:10.1073/pnas.90.18.8392 | RRID:CVCL_0063 | - |
| Cell line (*H. sapiens*) | T cell line SupT1 | *Smith et al., 1984*; PMID:6437672 | RRID:CVCL_1714 | - |
| Cell line (*H. sapiens*) | TZM-bl eGFP.OR3 IRES puro | This paper | - | Polyclonal TZM-bl cell line stably expressing eGFP.OR3 |
| Cell line (*H. sapiens*) | TZM-bl mScarlet.OR3 IRES puro | This paper | - | Polyclonal TZM-bl cell line stably expressing mScarlet.OR3 |
| Cell line (*H. sapiens*) | TZM-bl SNAP.OR3 IRES puro | This paper | - | Polyclonal TZM-bl cell line stably expressing SNAP.OR3 |
| Cell line (*H. sapiens*) | TZM-bl eBFP2.LMNB1 IRES BLR | This paper | - | Polyclonal TZM-bl cell line stably expressing eBFP2.LMNB1 |
| Cell line (*H. sapiens*) | TZM-bl eBFP2.LMNB1 IRES BLR eGFP.OR3 IRES puro | This paper | - | Polyclonal TZM-bl cell line stably expressing eBFP2.LMNB1 and eGFP.OR3 |
| Cell line (*H. sapiens*) | SupT1 eGFP.OR3 IRES puro | This paper | - | Polyclonal SupT1 cell line stably expressing eGFP.OR3 |
| Recombinant DNA reagent | pWPI eGFP.OR3 IRES puro | This paper | - | Lentiviral transfer plasmid containing eGFP.OR3 |
| Recombinant DNA reagent | pWPI mScarlet.OR3 IRES puro | This paper | - | Lentiviral transfer plasmid containing mScarlet.OR3 |
| Recombinant DNA reagent | pWPI SNAP.OR3 IRES puro | This paper | - | Lentiviral transfer plasmid containing SNAP.OR3 |
| Recombinant DNA reagent | pWPI eBFP2.LMNB1 IRES BLR | This paper | - | Lentiviral transfer plasmid containing eBFP2.LMNB1 |
| Recombinant DNA reagent | pWPI ANCH3 IRES puro | This paper | - | Lentiviral transfer plasmid containing ANCH3 |

*Continued on next page*

*Continued*

| Reagent type (species) or resource | Designation | Source or reference | Identifiers | Additional information |
|---|---|---|---|---|
| Recombinant DNA reagent | NNHIV | *Zila et al., 2021*; doi:10.1016/j.cell.2021.01.025 | - | Integration/transcription deficient HIV-1 proviral plasmid |
| Recombinant DNA reagent | NNHIV env(s) ANCH | This paper | - | Integration/transcription deficient HIV-1 proviral plasmid containing ANCH3 |
| Recombinant DNA reagent | NLC4-3 env(s) ANCH | This paper | - | HIV-1 proviral plasmid containing ANCH3 |
| Recombinant DNA reagent | Vpr-IN.SNAP | This paper | - | Expression of labelled IN |
| Recombinant DNA reagent | Vpr-IN$_{D64N/D116N}$.SNAP | This paper | - | Expression of labelled IN |
| Recombinant DNA reagent | Vpr-IN.eGFP | *Albanese et al., 2008*; doi:10.1371/journal.pone.0002413 | - | Expression of labelled IN |
| Recombinant DNA reagent | Vpr-IN$_{D64N/D116N}$.eGFP | other | - | Expression of labelled IN; D. Bejarano (University Hospital Heidelberg) |
| Recombinant DNA reagent | Vpr-IN.mScarlet | *Zila et al., 2021*; doi:10.1016/j.cell.2021.01.025 | - | Expression of labelled IN |
| Recombinant DNA reagent | Vpr-IN$_{D64N/D116N}$.mScarlet | *Zila et al., 2021*; doi:10.1016/j.cell.2021.01.025 | - | Expression of labelled IN |
| Recombinant DNA reagent | psPAX2 | Addgene | RRID:Addgene_35002 | Lentiviral packaging vector; D. Trono (EPFL, Lausanne, Switzerland) |
| Recombinant DNA reagent | pCMV-VSV-G | Addgene | RRID:Addgene_8454 | Expression of VSV-G; B. Weinberg (Whitehead Institute, MA, USA) |
| Recombinant DNA reagent | ANCHOR system | NeoVirTech (France) | - | http://www.neovirtech.com |
| Commercial assay, kit | Click-iT EdU Alexa Fluor 647 Imaging kit | Thermo Fisher Scientific | Cat#:C10340 | - |
| Commercial assay, kit | InviTrap Spin Universal RNA Mini kit | Stratec | Cat#:1060100300 | - |
| Software, algorithm | Fiji 1.53 c | *Schindelin et al., 2012*; doi:10.1038/nmeth.2019 | RRID:SCR_002285 | General image analysis |
| Software, algorithm | Icy 2.0.3.0 | *de Chaumont et al., 2012*; doi:10.1038/nmeth.2075 | RRID:SCR_010587 | Intensity quantification, Correlation |
| Software, algorithm | Imspector 16.1.6905 | Abberior Instruments | RRID:SCR_015249 | STED data acquisition and deconvolution |
| Software, algorithm | Prism 5.01 | GraphPad Software Inc | RRID:SCR_002798 | Visualization and Plotting |
| Software, algorithm | Matplotlib 3.1.3 | *Hunter, 2007*; doi:10.1109/MCSE.2007.55 | RRID:SCR_008624 | Visualization and Plotting |
| Software, algorithm | Seaborn 0.10.0 | *Waskom et al., 2020*; doi:10.5281/zenodo.3629446 | RRID:SCR_018132 | Visualization and Plotting |
| Software, algorithm | CSBdeep 0.4.1 | *Weigert et al., 2018*; doi:10.1038/s41592-018-0216-7 | - | Content-aware image restoration |
| Software, algorithm | Volocity 6.3 | Perkin Elmer | RRID:SCR_002668 | Data acquisition |
| Software, algorithm | IMOD 4.9.4 | *Kremer et al., 1996*; doi:10.1006/jsbi.1996.0013 | RRID:SCR_003297 | Tomogram Reconstruction |

*Continued on next page*

*Continued*

| Reagent type (species) or resource | Designation | Source or reference | Identifiers | Additional information |
|---|---|---|---|---|
| Software, algorithm | SerialEM 3.7.9 | *Mastronarde, 2005*; doi:10.1016/j.jsb.2005.07.007 | RRID:SCR_017293 | Tomogram acquisition, pre-correlation |
| Software, algorithm | ec-CLEM (Icy plugin) 1.0.1.5 | *Paul-Gilloteaux et al., 2017*; doi:10.1038/nmeth.4170 | - | Correlation |
| Software, algorithm | Amira-Avizo Software 2019.3 | Thermo Fisher Scientific | RRID:SCR_007353 | Visualization and rendering |
| Software, algorithm | PyMol 1.3 | Schrodinger LLC | RRID:SCR_000305 | Visualization and rendering |

## List of primers

| Primer | Sequence |
|---|---|
| Linearize NL4-3/NNHIV fw | CAGTTTTAATTGTGGAGGGG |
| Linearize NL4-3/NNHIV rv | ttaAGGTACCCCATAATAGAC |
| SNAP_Bam_fw | ccgcgcgggatccagggatggacaaagactgcgaaatg |
| SNAP_Not_rv | gccgcccgcggccgctttacagcccaggcttgcccagtct |
| eBFP2-LMNB1-10 into pWPI_BLR fw | tttccgatcacgagactagcctcgagg tttGCCACCATGGTGAGCAAG |
| eBFP2-LMNB1-10 into pWPI_BLR rv | tttactagtacgcgtgcgatcgccccggggg CTACATAATTGCACAGCTTCTATTGG |
| U1a Fwd primer | ACATCAAGCAGCCATGCAAAA |
| U1a Rev primer | CAGAATGGGATAGATTGCATCCA |
| U1a probe | AAGAGACCATCAATGAGGAA |
| Nuc1b Fwd primer | CGTCTGTTGTGTGACTCTGGTAACT |
| Nuc1b Rev primer | CACTGCTAGACATTTTCCACACTGA |
| Nuc1b probe | ATCCCTCAGACCCTTT |
| AluI (first round Alu PCR) | TCCCAGCTACTGGGGAGGCTGAGG |
| LM667 (first round Alu PCR) | ATGCCACGTAAGCGAAACTCTG GCTAACTAGGGAACCCACTG |
| λT (second round Alu qPCR) | ATGCCACGTAAGCGAAACT |
| LR (second round Alu qPCR) | TCCACACTGACTAAAAGGGTCTGA |
| ZXF-P (probe; second round Alu qPCR) | TGTGACTCTGGTAACTAGAGATCCCTCAGACCC |

## Plasmids

Plasmids were cloned using standard molecular biology techniques and verified by commercial Sanger sequencing (Eurofins Genomics, Germany). Gibson assembly was performed using the NEB HiFi Mastermix (New England Biolabs, USA) and 30 bp overlap regions. PCR was performed using Q5 High-Fidelity DNA Polymerase (New England Biolabs) according to manufacturer's instructions with primers purchased from Eurofins Genomics. *E. coli* DH5α and Stbl2 (*Trinh et al., 1994*, p. 2) (Thermo Fisher Scientific, USA) were used for amplification of standard plasmids or LTR containing plasmids, respectively.

## Derivatives of pNL4-3 and pNNHIV harboring ANCH1000 within the env gene

To facilitate the cloning procedure, EcoRI/XhoI fragments comprising the env region of HIV were subcloned from pNLC4-3 (*Bohne and Kräusslich, 2004*) and its non-replication competent derivative pNNHIV (*Zila et al., 2021*) into pcDNA3.1(+) (Thermo Fisher Scientific). These constructs were PCR

linearized, deleting a ~ 1000 bp region (nt 130–1113) within the *env* coding sequence. The ANCH3 1000 bp sequence was PCR amplified from pANCH3 (NeoVirTech, France), introducing a stop codon directly upstream of ANCH3, and transferred into the linearized vector fragments using Gibson assembly. The modified fragments were transferred into pNL4-3 or pNNHIV backbones using EcoRI/XhoI.

## Vpr-IN.SNAP and Vpr-IN$_{D64N/D116N}$.SNAP

The SNAP-tag gene was PCR amplified from pSNAP-tag(m) (Addgene #101135) and cloned into pVpr-IN.eGFP (*Albanese et al., 2008*) using BamHI/NotI, substituting the eGFP gene for the SNAP-tag coding region. To generate the Vpr-IN$_{D64N/D116N}$.SNAP mutant, the IN$_{D64N/D116N}$ from Vpr-IN$_{D64N/D116N}$.eGFP (gift from D. A. Bejarano) was cloned into Vpr-IN.SNAP using BamHI/NotI.

## Lentiviral transfer vectors harboring the ANCH sequence and eGFP. OR3, SNAP.OR3, mScarlet.OR3 and eBFP2.LMNB1 coding sequences

The ANCH3 1000 bp sequence was PCR amplified from pANCH3 (NeoVirTech) and cloned by Gibson assembly into pWPI IRES puro (*Trotard et al., 2016*) linearized with NotI. The eGFP.OR3 gene was PCR amplified from peGFP-OR3 (NeoVirTech) and transferred via Gibson assembly into PmeI/BamHI linearized pWPI IRES puro to create the expression cassette EF1-alpha eGFP.OR3 IRES puro. The SNAP gene was amplified from pVpr.IN.SNAP and the mScarlet (WT) (*Bindels et al., 2017*) gene from the mScarlet C1 vector (Addgene #85042) and placed N-terminal to OR3 into PCR linearized pWPI EF1-alpha OR3 IRES puro backbone by Gibson assembly, substituting eGFP. The eBFP2.LMNB1 gene was amplified from pEBFP2-LaminB1-10 (Addgene #55244) and transferred via Gibson assembly into PmeI/BamHI linearized pWPI IRES BLR (*Trotard et al., 2016*).

## Cell culture

HEK293T (*Pear et al., 1993*) (RRID:CVCL_0063), HeLa TZM-bl (*Wei et al., 2002*) (RRID:CVCL_B478), and SupT1 (*Smith et al., 1984*) (RRID:CVCL_1714) cell lines were authenticated using STR profiling (Eurofins Genomics) and monitored for mycoplasma contamination using the MycoAlert mycoplasma detection kit (Lonza Rockland, USA). Cells were cultured at 37°C and 5% $CO_2$ in Dulbecco's Modified Eagle's Medium (DMEM; Thermo Fisher Scientific) containing 4.5 g l$^{-1}$ D-glucose and L-glutamine supplemented with 10% fetal calf serum (FCS; Sigma Aldrich, USA), 100 U ml$^{-1}$ penicillin and 100 µg ml$^{-1}$ streptomycin (PAN Biotech, Germany) (adherent cell lines) or in RPMI 1640 (Thermo Fisher Scientific) containing L-glutamine supplemented with 10% FCS, 100 U ml$^{-1}$ penicillin and 100 µg ml$^{-1}$ streptomycin (SupT1 cells). Primary CD4$^+$ T cells were cultured in RPMI 1640 containing L-glutamine supplemented with 10% heat-inactivated FCS, 100 U ml$^{-1}$ penicillin and 100 µg ml$^{-1}$ streptomycin. Monocyte-derived macrophages (MDM) were cultured in RPMI 1640 containing 10% heat-inactivated FCS, 100 U ml$^{-1}$ penicillin, 100 µg ml$^{-1}$ streptomycin and 5% human AB serum (Sigma Aldrich).

## Isolation of primary cells

CD4$^+$ T cells were enriched from blood of healthy donors using RosetteSep Human CD4$^+$ T cell enrichment cocktail (Stemcell Technologies, Canada) according to the manufacturer's instructions followed by Ficoll density gradient centrifugation. Subsequently, cells were activated using human T-Activator CD3/CD28 Dynabeads (Thermo Fisher Scientific) and 90 U/ml IL-2 for 48–72 hr. MDMs were isolated from buffy coats of healthy blood donors as described previously (*Bejarano et al., 2019*).

## Generation of cell lines

Lentiviral vector particles were produced by co-transfection of packaging plasmid psPAX2 (Addgene #12260), the respective lentiviral transfer vector pWPI, the envelope expression plasmid pCMV-VSV-G (Addgene #8454) and pAdvantage (Promega, USA) in a ratio of 1.5: 1.0: 0.5: 0.2 µg into HEK293T cells (4 × 10$^5$ cells/well seeded the day before in six well plates) using polyethylenimine (PEI; 3 µl of 1 mg/ml PEI per µg DNA). At 48 hr post transfection, the tissue culture supernatant was harvested and filtered through 0.45 µm mixed cellulose ester (MCE) filters. SupT1 (1 ml of freshly 1:4 diluted cells) or TZM-bl (5 × 10$^4$ cells/well seeded the day before in 12 well plates) cells were

transduced using 50–500 µl supernatant. At 48 hr post transduction, selection with 1 µg/ml puromy-cin or 5 µg/ml blasticidin was initiated. For transduction of MDM, lentiviral vectors were produced with Vpx$_{mac239}$ (*Bejarano et al., 2018*) by calcium phosphate transfection of packaging plasmid pΔR8.9 NSDP (*Pertel et al., 2011*), containing a Vpx interaction motif in Gag, pWPI eGFP.OR3 IRES puro, Vpx expression plasmid pcDNA.Vpx$_{mac239}$ (*Sunseri et al., 2011*) and pCMV-VSV-G at a ratio of 1.33: 1.00: 0.17: 0.33 µg (68 µg / T175 flask). MDM were differentiated in human AB serum (Sigma Aldrich) from monocytes (*Bejarano et al., 2019*) in 15-well µ-Slides Angiogenesis (ibidi, Germany) for 10 days and transduction was performed 2 days prior to infection.

Production of viral particle stocks pNLC4-3 or pNNHIV ANCH, a Vpr-IN plasmid (Vpr-(SNAP/eGFP/mScarlet).IN or Vpr-(SNAP/eGFP/mScarlet).IN$_{D64N/D116N}$) and pCMV-VSV-G or pCAGGS.NL4-3-Xba (*Bozek et al., 2012*) were transfected in a ratio of 7.7: 1.3: 1.0 µg into HEK293T cells using calcium phosphate (70 µg / T175 flask). Medium was changed at 6–8 hr and cells were further incubated for 48 hr. Supernatant was harvested and filtered through 0.45 µm MCE before ultracentrifugation through a 20% (w/w) sucrose cushion (2 hr, 107,000 g). Pellets were resuspended in phosphate-buffered saline (PBS) containing 10% FCS and 10 mM HEPES (pH 7.5), and stored in aliquots at - 80°C. Virus was quantified using the SYBR Green based Product Enhanced Reverse Transcription assay (SG-PERT) (*Pizzato et al., 2009*). MOI of infectious particles was determined by titration on TZM-bl cells and immunofluorescence staining against HIV CA at 48 h p.i. The proportion of positive cells was counted in >10 randomly selected fields of view.

## Labeling of SNAP-tagged virus and infection

$3.33 \times 10^3$ TZM-bl cells were seeded into 15-well µ-Slides Angiogenesis (ibidi) the day before infection. Stock solutions of SNAP-Cell TMR-Star or SNAP-Cell 647-SiR (New England Biolabs) in DMSO were diluted to 4 µM in complete medium, mixed 1:1 with IN.SNAP particles and incubated at 37°C for 30 min. Labeled particles were added to cells at 5–30 µUnits RT/cell in 50 µl. For VSV-G pseudo-typed pNL4-3 ANCH, 30 µUnits RT per TZM-bl cell corresponds to ~MOI six in TZM-bl cells. Infection of MDM was performed with NNHIV ANCH (50 µl, 120 µUnits RT/cell). Infection of suspension cells was performed with $2 \times 10^4$ cells per 15 well µ-Slide in 96-well v-bottom plates (40 µl; 30 µU RT/cell). For PF-3450074 (PF74; Sigma Aldrich) experiments in primary CD4$^+$ T cells, medium was changed at 5 or 22 hr to medium containing 15 µM PF74 or DMSO, for 1 hr before transfer to PEI coated (with 1 mg/ml PEI for 60 min) µ-Slides. Slides were incubated for 1 hr for cell attachment prior to fixation. Efavirenz (EFV; Sigma Aldrich), Raltegravir (Ral; AIDS Research and Reference Reagent Program, Division AIDS, NIAID) and Azidothymidine (AZT) were added at time of infection. Flavopiridol (Sigma Aldrich) and 5,6-dichloro-1-beta-D-ribofuranosylbenzimidazole (DRB; Sigma Aldrich) were added 8 hr prior to fixation or RNA extraction. 10 µM EdU (Thermo Fisher Scientific) and 6 µM APC (Sigma Aldrich) were added at the time of infection.

## Fixation immunofluorescence staining and EdU click-labeling

Samples were washed with PBS and fixed (15 min, 4% PFA), washed again three times using PBS, permeabilized with 0.5% Triton X-100 (TX-100) for 10 or 20 min and washed again. In indicated experiments, cells were extracted using ice-cold methanol for 10 min. Afterwards, cells were washed two times using 3% bovine serum albumin (BSA)/PBS and blocked for 30 min with 3% BSA. Primary antibody in 0.5% BSA was added for 1 hr at room temperature. After washing three times with 3% BSA/PBS, secondary antibody in 0.5% BSA was added for 1 hr at room temperature and samples were washed and stored in PBS. For EdU incorporation experiments, cells were click-labeled for 30 min at room temperature using the Click-iT EdU-Alexa Fluor 647 Imaging Kit (Thermo Fisher Scientific) according to the manufacturer's instructions.

## DNA fluorescent in situ hybridization (FISH)

Biotinylated HIV-1 FISH probes were prepared with the Nick Translation Kit (Roche, Germany) according to the manufacturer's instructions. Probes were purified with Illustra Microspin G-25 columns (GE Healthcare, UK) according to the manufacturer's instructions and ethanol precipitated with human Cot-1 DNA (Roche, Germany) and herring sperm DNA (Sigma Aldrich). Probes were resuspended in 10 µL formamide, incubated at 37°C for 15–20 min and 10 µL of 20% dextran/4X saline-sodium citrate (SSC) buffer was added.

$1.25 \times 10^4$ TZM-bl cells/well were seeded on PEI (1 mg/ml) coated glass cover slips and infected with VSV-G pseudotyped IN.SNAP.SiR labeled NNHIV ANCH (30 µU/cell). At 24 hr, cells were fixed for 10 min with 4% PFA/PBS, permeabilized with 0.5% TX-100/0.1% Tween/PBS for 10 min at room temperature, and washed in 0.1% Tween/PBS. Following 30 min blocking in 4% BSA/PBS, cells were incubated with rabbit anti-GFP antibody (ab6556; Abcam, UK), diluted (1:2000) in 1% BSA/PBS overnight at 4°C. Cells were washed in 0.1% Tween/PBS and incubated with secondary Alexa Fluor antibody (Thermo Fisher Scientific) for 1 hr at room temperature. Cells were fixed for 10 min with 0.5 mM ethylene glycol bis(succinimidyl succinate) (EGS)/PBS, washed in 0.1% Tween/PBS and permeabilized with 0.5% TX-100/0.5% saponin/PBS for 10 min. Cells were incubated for 45 min in 20% glycerol/PBS and subjected to four glycerol/liquid $N_2$ freeze-thaw cycles. Samples were rinsed, incubated in 0.1 M HCl for 10 min, equilibrated in 2X SSC for 20 min and left in hybridization buffer (50% Formamide/2X SSC) for 30 min. Samples were then washed in PBS, treated with 0.01 N HCl/0.002% pepsin (3 min, 37°C) and quenched by addition of 1X PBS/1 M $MgCl_2$. Fixation with 4% PFA/PBS and PBS wash was followed by treatment with 100 µg/ml RNase A (PureLink, Invitrogen, USA) in 2x SSC for 30 min at 37°C, washing and overnight storage in hybridization buffer.

One to 10 µL of heat-denatured FISH probe (7 min at 95°C) was loaded onto glass slides covered with coverslips coated with prepared cells. Slides were sealed in a metal chamber heated at 80°C for 7 min, and incubated for 48 hr at 37°C. Samples were washed in 2X SSC at 37°C, followed by 3 washes in 0.5 X SSC at 56°C. FISH detection was performed using anti-biotin antibody (SA-HRP) and a FITC/Cy5 coupled secondary antibody with the TSA Plus system (Perkin Elmer, USA). Coverslips were stained with Hoechst, mounted on glass slides and imaged using the Nikon/Andor SDCM system described below.

## Confocal microscopy

Spinning disc confocal microscopy (SDCM) was performed on an inverted Nikon Eclipse Ti2 (Nikon, Japan) microscope equipped with a Yokogawa CSU-W1 Spinning Disk Unit (Andor, Oxford Instruments, United Kingdom) and an incubation chamber (37°C, 5% $CO_2$). Imaging was performed using a 100 × oil immersion objective (Nikon CFI Apochromat TIRF 100X Oil NA 1.49) and either single or dual-channel EMCCD camera setup (ANDOR iXon DU-888) recording the eBFP2 (405/420–460), eGFP (488/510–540 nm), mScarlet (568/589–625 nm) and SiR (647/665–705 nm) channels with a pixel size of 0.13 µm. 3D stacks of 10–30 randomly chosen positions were automatically recorded with a z-spacing of 0.3–0.5 µm using the Nikon Imaging Software Elements v5.02. For CLEM experiments a Perkin Elmer Ultra VIEW VoX 3D spinning disk confocal microscope (Perkin Elmer, United States) with a 100 x oil immersion objective (NA 1.4; Perkin Elmer) was used.

## Live cell imaging

Medium was exchanged for 50 µl imaging medium containing FluoroBrite DMEM (Thermo Fisher Scientific), 10% FCS, 4 mM GlutaMAX (Gibco Life Technologies), 2 mM sodium pyruvate (Gibco Life Technologies), 20 mM HEPES pH 7.4, 100 U/ml Penicillin and 100 µg/ml Streptomycin (PAN-Biotech). Samples were transferred to the SDCM setup described above. 3D stacks were recorded up to 24 hr (time interval of 3–30 min, z-spacing 0.5 µm). Data from *Figure 1g,f* and *Figure 4h* was fit to a four-parametric logistic population growth model with variable slope using Prism 5.01 (Graphpad).

$$y = a + (b - a) * (1 + 10^{(\text{Log}(t1/2 - t) - n)})^{-1}$$

with y = normalized eGFP.OR3 events per cell, a = Y value at the bottom plateau, b = Y value at the top plateau, t = h p.i., $t_{1/2}$ = time at half maximal signal and n = slope factor.

## STED microscopy

$3.33 \times 10^3$ SNAP.OR3 expressing cells were seeded on 15-well µ-Slides Angiogenesis (ibidi) and infected as described above. Prior to fixation, cells were incubated with 2 µM SNAP-Cell 647-SiR (New England Biolabs) for 30 min at 37°C, washed three times and fixed with 4% PFA (15 min). STED microscopy was performed using a 775 nm STED system (Abberior Instruments GmbH, Germany) equipped with a 100 x oil immersion objective (NA 1.4; Olympus UPlanSApo). STED Images were acquired using the 590 and 640 nm excitation laser lines while the 405 and 488 laser lines were

acquired in confocal mode. Nominal STED laser power was set to 80% of the maximal power (1250 mW) with 20 μs pixel dwell time and 15 nm pixel size. STED Images were linearly deconvolved with a Lorentzian function (FWHM 50 nm) using the Richardson-Lucy algorithm and the software Imspector (Abberior Instruments GmbH).

## Image analysis and data visualization

The images were filtered in Fiji/ImageJ (*Schindelin et al., 2012*) with a mean filter (kernel size: 0.26 × 0.26 μm) to reduce noise. For visualization some of the low signal-to-noise 3D movies were denoised using content-aware image restoration (CARE) (*Weigert et al., 2018*) as indicated. Convolutional neuronal networks were trained with a set of fixed cell images recorded with high laser power/long camera exposure (ground truth) and low laser power/short camera exposure (noisy training input) using the Python toolbox CSBdeep (*Weigert et al., 2018*). The model was then applied to reconstruct raw movies in Fiji using the CSBdeep CARE plugin. Quantification of fluorescent spot intensities was performed using Icy (*de Chaumont et al., 2012*). Raw 3D stacks were used to detect volumes of IN objects using the spot detector plugin. Methanol-induced shrinkage of cells in z orientation and infection-induced invaginations of the nuclear envelope renders automated nuclei detection difficult. To ensure that only truly nuclear objects (and not still NE-associated ones) were classified as such, they were manually curated and ambiguous particles were excluded. Objects displaying positive signals in the lamin channel were classified as nuclear envelope (NE) associated in the TZM-bl experiments. For measurements in primary CD4[+] T cells, we applied a more stringent classification, manually excluding objects that did not colocalize with lamin in the major part of the signal or were localized above/below the focal planes of the nucleus. Cell-specific local background was subtracted for CA quantification. For quantification of CPSF6 intensities, the diffusive nuclear expression level of the cell was measured (using a ROI without punctae) and intensities of CPSF6 accumulations at IN objects were normalized to the expression level of the cell. Nuclear OR3 punctae were counted if their intensity was $\geq$20% above the diffuse nuclear expression level of the respective cell. Colocalization was scored when the pixel areas of the respective fluorescent spots (partially) overlapped. Tracking was performed using the Fiji plugin Manual Tracking. Fiji standard 'Fire' lookup table (LUT) was used for visualization of single channel images. Statistical tests were performed using Prism v5.01 (GraphPad Software Inc, USA). Data were plotted using Prism v5.01 or the Python statistical data visualization package matplotlib v3.1.3 (*Hunter, 2007*) and seaborn v0.10.0 (*Waskom et al., 2020*). Graphs show mean with error bars defined in the figure legends.

## Quantification of RT products

Particle preparations filtered through 0.45 μm CME were treated with 15 U/mL DNaseI (Sigma Aldrich)/10 mM MgCl$_2$ for 3–4 hr at 37°C prior to ultracentrifugation. 5 × 10$^4$ TZM-bl cells were seeded into 24-well plates and infected the following day using 10–30 μU RT/cell. At the indicated h p.i. cells were washed, scraped and lysed using 50 μl of 10 mM Tris-HCl pH 9.0, 0.1% TX-100, 400 μg/mL proteinase K (Thermo Fisher Scientific) at 55°C overnight. Proteinase K was inactivated at 95°C for 10 min and lysates were stored at −20°C. Alternatively, DNA was purified from cells using the DNeasy Blood and Tissue Kit (Qiagen, Germany) according to the manufacturer's instructions. Lysates were directly used as input for ddPCR; purified DNA was prediluted to ~20 ng/μl. For *gag* cDNA detection, this input was additionally diluted 1:20 to prevent saturation. ddPCR was performed using the QX200 droplet generator/reader (BioRad, USA) and analyzed using QuantaSoft v1.7.4 (BioRad) as described earlier (*Bejarano et al., 2018*; *Zila et al., 2019*).

Quantification of viral RNA transcripts eBFP2.LMNB1 and eGFP.OR3 expressing TZM-bl cells were infected with VSV-G pseudotyped NL4-3 ANCH (5 μUnits RT/cell; MOI ~ 1) for 55 hr. Twenty μm RAL was added at the time of infection. 8 hr prior to RNA extraction and purification using the Invitrap spin universal RNA mini kit (Stratec biomedical, Germany), medium was changed for medium containing 1–25 μM flavopiridol (Sigma Aldrich) or 1–25 μM 5,6-dichloro-1-beta-D-ribofuranosylbenzimidazole (DRB; Sigma Aldrich) as indicated. Quantitative reverse transcription PCR was performed as previously described (*Marini et al., 2015*). Briefly, messenger RNA (mRNA) levels were quantified by TaqMan quantitative RT-PCR (qRT-PCR). First, the RT reaction was performed using M-MLV RT (Invitrogen) and a random primer set (Invitrogen, cat. no.: 48190011), followed by qPCR using HIV-1 primers and probes (specific for transcription of the first nucleosome nuc1a or

gag/u1a) and the housekeeping genes 18S and GAPDH (both containing VIC/TAMRA fluorescent probe; Applied Biosystems, USA) as controls (see list of primers).

## Quantification of integrated proviral DNA using Alu PCR

A total of $2 \times 10^5$ SupT1 cells were infected using VSV-G pseudotyped HIV-1 NL4-3 ANCH (10 µU RT/cell) for 24 or 48 hr. Twenty µm Ral or 20 µm EFV was added at the time of infection. Cells were washed, lysed and genomic DNA was extracted using the DNeasy Blood and Tissue Kit (Qiagen) according to the manufacturer's instructions. Nested Alu-LTR PCR was performed as described before (*Tan et al., 2006*). Briefly, Alu-LTR fragments were amplified starting from 100 ng of genomic DNA. The product of the first amplification was diluted 1:50 in $H_2O$ and amplified by qPCR. The B13 region of the housekeeping gene lamin B2 was amplified from 10 ng of genomic DNA for normalization (*Livak and Schmittgen, 2001*). The copy number (Log10) of integrated HIV-1 DNA per million cells was calculated using a standard curve obtained by serially diluting DNA from HIV-1 infected and sorted p24$^+$ cells with DNA from uninfected cells as described before (*Shytaj et al., 2020*).

## CLEM sample preparation

$1.2 \times 10^5$ TZM-bl mScarlet.OR3 cells were grown on 3 mm sapphire discs in a 35 mm glass-bottom dish (MatTek, USA). Cells were infected with VSV-G pseudotyped IN.SNAP.SiR labeled NNHIV ANCH (30 µU RT/cell) and incubated for 24 hr at 37°C. Subsequently, cells were cryo-immobilized by high pressure freezing (HPF) (HPM010; Abra Fluid, Switzerland) and transferred to freeze-substitution (FS) medium (0.1% uranyl acetate, 2.3% methanol and 1% $H_2O$ in Acetone) tempered at −90°C. Freeze-substituted samples were embedded in Lowicryl HM20 resin (Polysciences, USA) inside a FS device (AFS2, Leica, Germany) equipped with a robotic solution handler (FSP, Leica). FS and embedding into Lowicryl resin was performed according to *Kukulski et al., 2011* with modifications (*Zila et al., 2021*). Temperature was raised to −45°C at 7.5 °C/hr. Samples were washed with acetone (4 × 25 min) and infiltrated with increasing concentrations of Lowicryl in acetone (25, 50% and 75%; 3 hr each), while raising temperature to −25°C (3.3 °C /hr). The acetone-resin mixture was replaced by Lowicryl HM20 for 1 hr and the resin was exchanged three times (3, 5 and 12 hr). Samples were polymerized under UV light for 24 hr at −25°C. Polymerization continued for an additional 24 hr while the temperature was raised to 20°C at 3.7 °C/hr.

## CLEM and electron tomography

Thick resin sections (250 nm) were cut using a microtome (EM UC7, Leica) and placed on a slot (1 × 2 mm) EM copper grid covered with a formvar film (FF2010-Cu, Electron Microscopy Sciences, USA). Sections were covered by 0.1 µm TetraSpeck microsphere fiducials (Thermo Fisher Scientific). Nuclear regions were stained with 50 µg/ml Hoechst (Thermo Fisher Scientific). For SDCM, grids were transferred to 25 mm glass coverslips mounted in a water-filled ring holder (Attofluor cell chamber, Thermo Fisher Scientific). Z stacks of cell sections were acquired using the PerkinElmer UltraVIEW VoX 3D Spinning-disc Confocal Microscope described above (z-spacing 200 nm). To identify mScarlet.OR3 and IN.SNAP.SiR signals in cell sections, images were visually examined using Fiji (*Schindelin et al., 2012*). Subsequently, both sides of EM grids were decorated with 15 nm protein-A gold particles for tomogram alignment and contrasted with 3% uranyl acetate (in 70% methanol) and lead citrate. Individual grids were placed in a high-tilt holder (Fischione Model 2040) and loaded into the Tecnai TF20 (FEI, Eindhoven, Netherlands) electron microscope operated at 200 kV, equipped with a field emission gun and a 4 K by 4 K pixel Eagle CCD camera (FEI, USA). To identify positions for ET, a full grid map was acquired using SerialEM (*Mastronarde, 2005*) and acquired electron micrographs were pre-correlated with imported SDCM images in SerialEM using fiducials as landmark points (*Schorb et al., 2017*). Single-axis electron tomograms of selected regions were then carried out. Tomographic tilt ranges were from −60° to 60° with an angular increment of 1° and pixel size 1.13 nm. Alignments and 3D reconstructions of tomograms were done using IMOD (*Kremer et al., 1996*). For high precision post-correlation, tomograms of cell sections were acquired at lower magnification with 4° increment and 6.3 nm pixel size. Post-correlation was performed using the eC-CLEM plugin (*Paul-Gilloteaux et al., 2017*) in Icy (*de Chaumont et al., 2012*). The length and diameter of capsids were measured in IMOD (*Kremer et al., 1996*). Segmentation and isosurface rendering were performed in Amira (Thermo Scientific).

## Acknowledgements

The ANCHOR system is developed by and commercially available from NeoVirTech (France, http://www.neovirtech.com). We gratefully acknowledge Harald Wodrich (University of Bordeaux, France) and Franck Gallardo (NeoVirTech) for helpful discussions. We thank David Bejarano (University Hospital Heidelberg, Germany) for providing pVpr-IN$_{D64N/D116N}$.eGFP. EBFP2-LaminB1-10 was a gift from Michael Davidson (Addgene plasmid # 55244), pSNAP-tag (m) Vector from New England Biolabs and Ana Egana (Addgene plasmid # 101135), pmScarlet_C1 from Dorus Gadella (Addgene plasmid # 85042), psPAX2 from Didier Trono (Addgene plasmid #12260) and pCMV-VSV-G was from Bob Weinberg (Addgene plasmid #8454). The following reagent was obtained through the NIH HIV Reagent Program, Division of AIDS, NIAID, NIH: Anti-Human Immunodeficiency Virus 1 (HIV-1) p24 Monoclonal (71-31), ARP-530, contributed by Dr. Susan Zolla-Pazner. We thank Anke-Mareil Heuser and Vera Sonntag-Buck for excellent technical assistance. We acknowledge the microscopy support from the Infectious Diseases Imaging Platform (IDIP) of the Center for Integrative Infectious Disease Research, Heidelberg. This work was funded by the Deutsche Forschungsgemeinschaft (DFG, German Research Foundation) – Projektnummer 240245660 – SFB 1129 project 5 (H-GK), project 6 (BM), project 20 (ML) and by the TTU HIV in the DZIF (VL, ML, H-GK).

## Additional information

### Funding

| Funder | Grant reference number | Author |
|---|---|---|
| Deutsche Forschungsgemeinschaft | Projektnummer 240245660 - SFB 1129 project 5 | Thorsten G Müller Vojtech Zila Kyra Peters Hans-Georg Kräusslich |
| Deutsche Forschungsgemeinschaft | Projektnummer 240245660 - SFB 1129 project 6 | Sandra Schifferdecker Barbara Müller |
| Deutsche Forschungsgemeinschaft | Projektnummer 240245660 - SFB 1129 project 20 | Mia Stanic Bojana Lucic Marina Lusic |
| Deutsches Zentrum für Infektionsforschung | TTU HIV | Vibor Laketa Marina Lusic Hans-Georg Kräusslich |

The funders had no role in study design, data collection and interpretation, or the decision to submit the work for publication.

### Author contributions

Thorsten G Müller, Conceptualization, Formal analysis, Investigation, Visualization, Methodology, Writing - original draft, Writing - review and editing; Vojtech Zila, Formal analysis, Investigation, Visualization, Methodology, Writing - original draft, Writing - review and editing; Kyra Peters, Sandra Schifferdecker, Mia Stanic, Bojana Lucic, Vibor Laketa, Formal analysis, Investigation, Writing - review and editing; Marina Lusic, Supervision, Funding acquisition, Writing - review and editing; Barbara Müller, Supervision, Funding acquisition, Visualization, Writing - original draft, Writing - review and editing; Hans-Georg Kräusslich, Conceptualization, Supervision, Funding acquisition, Writing - original draft, Writing - review and editing

### Author ORCIDs

Thorsten G Müller ⓘD https://orcid.org/0000-0002-4197-6224
Vojtech Zila ⓘD https://orcid.org/0000-0003-2032-3600
Sandra Schifferdecker ⓘD https://orcid.org/0000-0003-1473-5748
Mia Stanic ⓘD https://orcid.org/0000-0003-2149-2916
Bojana Lucic ⓘD https://orcid.org/0000-0001-5300-2489
Vibor Laketa ⓘD http://orcid.org/0000-0002-9472-2738
Marina Lusic ⓘD http://orcid.org/0000-0002-0120-3569

Barbara Müller 🆔 http://orcid.org/0000-0001-5726-5585
Hans-Georg Kräusslich 🆔 https://orcid.org/0000-0002-8756-329X

**Decision letter and Author response**
Decision letter https://doi.org/10.7554/eLife.64776.sa1
Author response https://doi.org/10.7554/eLife.64776.sa2

## Additional files

**Supplementary files**
• Transparent reporting form

**Data availability**

All data generated or analysed during this study are included in the manuscript and supporting files. Source data files have been provided for Figure 1f,h, Figure 1 figure supplement 1c-d, Figure 1 figure supplement 3a, Figure 2a-b, Figure 2e-f, Figure 3b-c, Figure 4c,g,h,j-k,o, Figure 4 - figure supplement 1a-b, Figure 7c,e-f, Figure 7 - figure supplement 1b, Figure 7 - figure supplement 2a-b. Materials involving the ANCHOR system are MTA-restricted and commercially available from NeoVirTech (France).

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
