## [Decision Letter]

**Acceptance summary:**

The authors use a variety of complementary approaches to visualize and characterize events in the first half of the HIV life cycle. The high quality data are in line with several other recent studies indicating that reverse transcription completes in the nucleus, that intact/nearly intact cores are imported into the nucleus, and that nuclear uncoating likely occurs immediately prior to integration. Most importantly, the results provide the best evidence to date that largely intact capsids can enter the nucleus of target cells during infection. These results are very different from the previous paradigms that held that reverse transcription and capsid uncoating occur almost exclusively in the cytoplasm.

**Decision letter after peer review:**

Thank you for submitting your article "HIV-1 uncoating by release of viral cDNA from capsid-like structures in the nucleus of infected cells" for consideration by *eLife*. Your article has been reviewed by 3 peer reviewers, and the evaluation has been overseen by a Reviewing Editor and Sara Sawyer as the Senior Editor. The following individual involved in review of your submission has agreed to reveal their identity: Edward M Campbell (Reviewer #3).

Essential revisions:

1. Epitope unmasking in the nucleus.

The authors argue that CA epitopes on nuclear capsids are normally masked by CPSF6, thereby impairing visualization in primary CD4^+^T cells, and that PF74 treatment can unmask the hidden epitopes (Figure 6). However, according to some publications 15µM PF74 destabilizes core-like structures in cells. The authors should therefore demonstrate that core-like structures can still exist in the nucleus at this PF74 dose. Alternatively, antibody recognition could instead reflect remodeling of the core-like viral structure to expose epitopes recognized by the antibody. Similarly, the authors mention that in macrophages, the nuclear CA signal is easier detected by IF than in T cells. Shouldn't CPSF6 shield CA epitopes similarly in both cells? Why is it possible to detect the intranuclear CA in macrophages without necessarily use PF74? Finally, PF74 is also known to inhibit Nup153 binding to CA. What controls were done to assure inhibition of Nup153 binding did not contribute to phenotypes?

2. A key conclusion of this study is that intact cores can enter the nucleus through nuclear pores. However, aphidicolin treatment does not necessarily completely block the division of all cells, and the authors should show that the analyzed cells cannot divide, for example by using EdU (which according the authors does not stain the DNA of Aph treated cells), although this may be technically complicated. Regardless, the complete block to cell division in the analyzed cell must be demonstrated to rule out core entry during mitotic nuclear breakdown.

---

## [Author Response]

Essential revisions:1. Epitope unmasking in the nucleus.The authors argue that CA epitopes on nuclear capsids are normally masked by CPSF6, thereby impairing visualization in primary CD4^+^T cells, and that PF74 treatment can unmask the hidden epitopes (Figure 6). However, according to some publications 15µM PF74 destabilizes core-like structures in cells. The authors should therefore demonstrate that core-like structures can still exist in the nucleus at this PF74 dose. Alternatively, antibody recognition could instead reflect remodeling of the core-like viral structure to expose epitopes recognized by the antibody. Similarly, the authors mention that in macrophages, the nuclear CA signal is easier detected by IF than in T cells. Shouldn't CPSF6 shield CA epitopes similarly in both cells? Why is it possible to detect the intranuclear CA in macrophages without necessarily use PF74? Finally, PF74 is also known to inhibit Nup153 binding to CA. What controls were done to assure inhibition of Nup153 binding did not contribute to phenotypes?

As stated in the review, we believe that CA epitope masking in the nucleus is a potential reason for not (or weakly) detecting CA on nuclear HIV-1 complexes by immunostaining. This may be overcome by various extraction methods and/or by PF74-treatment, which interferes with CPSF6-binding to the CA-lattice. We did not claim, however, that stable core-like structures are retained after PF74 treatment as implied in the review. Our data suggest that CA signal intensities on nuclear subviral complexes are similar to those on cytosolic complexes indicating that most CA molecules are retained on the HIV-1 subviral complex after nuclear entry. Our CLEM analysis showed cone-shaped and tubular core-like structures as well as apparent core remnants in the nucleus without PF74 treatment. Whether these structures remain intact after PF74 treatment was not analyzed, however, and is not part of our argument.

This leaves the important question whether the observed CA signal is primarily due to either unmasking of CA by CPSF6 removal or to CA epitope exposure due to remodeling of the capsid structure by PF74. To address this issue, we analyzed CPSF6 and CA-signals depending on PF74 concentration and time of treatment and compared signals on nuclear and cytosolic complexes. We also made use of a monoclonal antibody whose epitope is close to the CPSF6 binding site.

Additional experiments exploring the effect of extraction and PF74 treatment were performed with TZM-bl cells, SupT1 cells and primary CD4^+^ T cells. We show that:

i. Either PF74 treatment or methanol extraction led to detection of previously hidden CA signals on nuclear HIV-1 subviral complexes in TZM-bl cells. (new Figure 4a, b, f and Figure 4 — figure supplement 1b).

ii. Recognition and signal intensity of the nuclear CA-signal on HIV-1 subviral complexes was dependent on PF74 concentration and time of treatment (Figure 4g, h).

iii. The CA signal intensity remained unchanged on extracellular virions and cytosolic HIV-1 subviral complexes when treated with the same PF74 concentration in the same cells and experiment; this shows that PF74 treatment does not cause increased staining due to exposure of novel epitopes (Figure 4 — figure supplement 1a, b see also v).

iv. The intensities of CPSF6 immunostaining and CA immunostaining were inversely correlated (new Figure 4g)

v. Similar results as for the polyclonal antiserum were obtained for a monoclonal antibody with a known epitope in the vicinity of the CPSF6 binding region (Figure 4 — figure supplement 1c, d). This antibody efficiently bound to cytoplasmic complexes and glass-bound particles, indicating that the respective epitope is exposed in the context of the assembled capsid; no signal was observed on nuclear HIV-1 subviral complexes in the absence of PF74 extraction, however. Following PF74 treatment, CA signals were observed on nuclear subviral complexes as well and matched the intensity on cytosolic complexes (new Figure 4j, k). No difference in signal intensity was observed on cytosolic complexes on the other hand, indicating that this treatment does not alter exposure of the epitope.

vi. CA signals on nuclear subviral HIV-1 complexes were also analyzed in a T-cell line and primary CD4^+^ T cells: detection of nuclear CA signals required treatment with PF74 and methanol extraction (new Figure 7 — figure supplement 2) in this cell type, while no change in CA signal intensities was observed on cytosolic HIV-1 subviral complexes under the same conditions.

In summary, our data indicate that the CA signal on nuclear HIV-1 subviral complexes is largely hidden due to dense coating of the subviral complex with CPSF6. Weak and variable CA staining was reported for nuclear subviral complexes in different reports (Chen et al., 2016; Chin et al., 2015; Francis et al., 2020; Hulme et al., 2015; Peng et al., 2015; Zhou et al., 2011) and these differences may be due to differences in fixation and extraction methods. Experiments in MDM (Bejarano et al., 2018; Stultz et al., 2017) and THP-1 cells (Rensen et al., 2021) detecting strong nuclear CA signals were generally performed following cell extraction for copper-mediated click reaction. Thus, reliable detection of CA signals on nuclear HIV-1 subviral complexes by immunofluorescence appears to depend on removal of CPSF6 by PF74 and/or on cell extraction (e.g. by methanol). We currently do not know why different treatment intensities are required in different cell types; these observations may indicate cell-type specific differences in the architecture of the CPSF6 cluster and/or the presence of additional cell-type specific proteins. These aspects are now mentioned in the Discussion of the revised manuscript. Our findings are also consistent with Chin et al., 2015, who reported CA immunodetection on nuclear HIV-1 subviral complexes to be dependent on prior partial protease digestion of the sample.

We conclude that the majority of CA of the incoming HIV-1 capsid is retained on nuclear subviral complexes in all cell types analyzed; these observations are in line with very recent findings by us and others (Burdick, 2020; Zila, Margiotta et al., 2021). Prior failure to detect CA on nuclear subviral complexes by immunofluorescence appears to have been largely due to masked epitopes.

Regarding PF74-mediated inhibition of Nup153 binding, we want to emphasize that PF74 treatment in our study was done for short periods of time at later time points after infection. We thus analyzed HIV-1 subviral complexes that had already entered the nucleus and separated from the NPC prior to addition of PF74. Accordingly, the CA-Nup153 interaction or its inhibition was not analyzed in this study.

2. A key conclusion of this study is that intact cores can enter the nucleus through nuclear pores. However, aphidicolin treatment does not necessarily completely block the division of all cells, and the authors should show that the analyzed cells cannot divide, for example by using EdU (which according the authors does not stain the DNA of Aph treated cells), although this may be technically complicated. Regardless, the complete block to cell division in the analyzed cell must be demonstrated to rule out core entry during mitotic nuclear breakdown.

We agree with the reviewer that this is an important point that needs to be addressed. We have performed additional experiments to address this issue. Aphidicolin (APC) treated TZM-bl cells expressing eBFP2.LMNB1 to mark the nuclear envelope were infected and followed by live cell imaging of the blue channel for 12h p.i. At this time point, cells were briefly treated with PF74, fixed, immunostained and imaged. Fixed cells were subsequently re-identified in the live cell imaging video and image analysis and quantitation was performed exclusively for cells that had not undergone nuclear breakdown during the entire 12 h period. We clearly identified CA-positive nuclear HIV-1 subviral complexes in these cells that must have entered the nucleus through intact nuclear pores. The CA signal intensity was again similar to that on cytosolic or extracellular HIV-1 particles, validating our conclusion that largely intact capsids enter the nucleus through intact nuclear pores. The new data are presented as new Figure 4 — figure supplement 1e, f and new Figure 4l-o in the revised manuscript.